# Field-free spin-orbit torque switching assisted by in-plane unconventional spin torque in ultrathin [Pt/Co]N

Fen Xue[1] ✉, Shy-Jay Lin[2], Mingyuan Song[2], William Hwang [1], Christoph Klewe [3], Chien-Min Lee[2], Emrah Turgut[2], Padraic Shafer [3], Arturas Vailionis [4,5], Yen-Lin Huang [2], Wilman Tsai[6], Xinyu Bao[2] & Shan X. Wang [1,6] ✉

Electrical manipulation of magnetization without an external magnetic field is critical for the development of advanced non-volatile magnetic-memory technology that can achieve high memory density and low energy consumption. Several recent studies have revealed efficient out-of-plane spin-orbit torques (SOTs) in a variety of materials for field-free type-z SOT switching. Here, we report on the corresponding type-x configuration, showing significant in-plane unconventional spin polarizations from sputtered ultrathin [Pt/Co]N, which are either highly textured on single crystalline MgO substrates or randomly textured on SiO₂ coated Si substrates. The unconventional spin currents generated in the low-dimensional Co films result from the strong orbital magnetic moment, which has been observed by X-ray magnetic circular dichroism (XMCD) measurement. The x-polarized spin torque efficiency reaches up to −0.083 and favors complete field-free switching of CoFeB magnetized along the in-plane charge current direction. Micromagnetic simulations additionally demonstrate its lower switching current than type-y switching, especially in narrow current pulses. Our work provides additional pathways for electrical manipulation of spintronic devices in the pursuit of high-speed, high-density, and low-energy non-volatile memory.

Spin currents arising from charge currents in materials with strong spin-orbit coupling (SOC) have provided an efficient and ultrafast approach for magnetization control via spin-orbit torque (SOT) in magnetoresistive random-access memory (MRAM) and logic devices[1]. As an embedded non-volatile memory technology, SOT-MRAM exhibits great potential to compete with the last-level cache SRAM[2] due to its sub-ns switching speed, sub-1V switching voltage[3], and high endurance, which is achieved by a three-terminal bit cell to separate the read and write paths in contrast to spin-transfer-torque (STT)

MRAM. However, there are still several challenges for SOT-MRAM with respect to efficiency and density.

Without an external magnetic field, conventional SOTs generated by in-plane spin polarization $\sigma_y$ can only deterministically switch type-y magnetic tunnel junctions (MTJs). Here, type-y refers to systems where the magnetic easy axis of the switching layer is along $y$ direction, transverse to the current ($x$) direction; type-x and type-z are similarly defined without changing the current direction. Such type-y SOT-MRAM requires ~10–100× longer switching time and ~3× larger cell size

[1]Department of Electrical Engineering, Stanford University, Stanford, CA 94305, USA. [2]Taiwan Semiconductor Manufacturing Company, Hsinchu, Taiwan. [3]Advanced Light Source, Lawrence Berkeley National Laboratory, Berkeley, CA 94720, USA. [4]Stanford Nano Shared Facilities, Stanford University, Stanford, CA 94305, USA. [5]Department of Physics, Kaunas University of Technology, LT-51368 Kaunas, Lithuania. [6]Department of Materials Science and Engineering, Stanford University, Stanford, CA 94305, USA. ✉e-mail: fenx@stanford.edu; sxwang@stanford.edu

compared to type-x and type-z geometries[4]. Limited by the spin polarizations generated in heavy metals[5], deterministic SOT switching in type-x (type-z) requires symmetry breaking with a perpendicular (an in-plane) magnetic field/exchange bias[6,7], or an asymmetric lateral design[8]. Field-free switching is technologically important as the integration of external-field sources into nanodevices undesirably limits the scaling. The exchange coupling from the interface of antiferromagnetic (AFM) and ferromagnetic (FM) films can break the symmetry and make it promising in field-free type-z geometry[9], however, this exchange bias is not stable since it could be manipulated by current-induced SOTs[10]. Introducing lateral symmetry breaking into the structure[11] shows clear switching behavior, however, it is not ideal for wafer-level application due to the necessary homogeneity in magnetic and transport properties for mass production. Therefore, these "extrinsic" approaches are not favorable with respect to memory density, back-end-of-line (BEOL) compatibility, magnetic immunity, etc.

A promising alternative is to explore "intrinsic" material engineering such as breaking inversion symmetry of the SOC materials so as to generate unconventional spin polarizations $\sigma_x$ ($\sigma_z$) for in-plane (out-of-plane) SOTs. Several recent studies revealed out-of-plane SOTs due to broken spin conservation in the non-collinear spin configuration in materials like $L1_1$-ordered CuPt[12], and AFM materials like $Mn_2Au$[13], $Mn_3Sn$[14], and $MnPd_3$[15]. In these studies, a spin current flowing in the $z$-direction with an unconventional $z$-polarized spin orientation, $\sigma_z$, parallel to the spin-current direction, is induced by a charge current in the $x$-direction. The SOTs from $z$-polarized spins can then drive the field-free switching in the neighboring magnetic free layer with a magnetic easy-axis aligned along the $z$-direction (type-z). The corresponding structure, type-x, where the magnetic easy-axis aligned along the $x$-direction, has not been reported with field-free SOT switching favored by SOTs from unconventional spin polarizations $\sigma_x$, which is polarized in parallel with the magnetic easy-axis along the $x$-direction.

Governed by similar switching mechanisms and dynamic trajectories, type-x SOT switching scheme has the advantage of being faster than type-z SOT switching due to the presence of an out-of-plane demagnetization field[16], with five times lower switching current density than type-z while maintaining a reasonably high thermal stability factor $E/k_BT$[4], and benefits more from scaling (Supplementary Fig. 1). According to micromagnetic simulations of nanometer-scale elliptical MTJ cells[2], the polarized spins with ratio of $\sigma_x/\sigma_y = 0.017$ enable the field-free deterministic type-x SOT switching, while a much higher ratio of $\sigma_z/\sigma_y = 0.520$ is needed to deterministically switch type-z geometry, with a similar switching current of 2 ns as in the type-y geometry with pure conventional spin polarizations $\sigma_y$. Therefore, the field-free type-x SOT switching is much easier than the type-z in view of the unconventional spin torque efficiency required in deterministic switching. Motivated by the theoretical advantages mentioned above, this work focuses on the $x$-polarized spins generated in SOC materials and demonstrates field-free SOT switching in a type-x geometry.

In this work, we first report the giant unconventional spin torque efficiency corresponding to the $x$-polarized spins $\theta_{AD,x}$ from magnetron sputtered $[Pt(1)/Co(0.16)]_5$ films on MgO crystalline substrates, which is determined to be −0.083 while having a conventional spin conductivity $\sigma_y$ of ~3.4 × 10$^5$ $\hbar/2e$ $\Omega^{-1}$m$^{-1}$. Next, we demonstrate complete field-free switching of the in-plane CoFeB layer magnetized along the charge current direction via differential planar Hall effect (DPHE) measurements. Then, micromagnetic simulations are conducted to understand the dynamics of $x$-polarized spins assisting field-free type-x switching. Finally, we discuss the Co magnetization and thickness dependence of spin polarizations and elucidate the origin of unconventional spin polarizations generated in low-dimensional Co by X-ray magnetic circular dichroism (XMCD) experiments. Taken together, the CMOS-compatible SOT materials as reported here provide a promising approach for the electrical manipulation of spintronic devices such as SOT-MRAM.

## Results

### Giant unconventional SOT efficiencies in [Pt/Co]$_N$ stacks with ultrathin Co

In our $[Pt(1\,nm)/Co(0.159\,nm)]_5/Mg(2\,nm)/CoFeB(2.5\,nm)$ as-deposited film stacks grown on (100) MgO single crystals, the unconventional and conventional in-plane SOT efficiencies were found to be comparable. Here, $[Pt(1\,nm)/Co(0.159\,nm)]_5$ is denoted as SRT1, and the stack is abbreviated as SRT1/Mg/CFB. As shown in the schematic of MgO crystallized structures in Fig. 1a, we adopted three substrates with different crystalline plane orientations to grow the film stacks. The sputtered Pt films grow epitaxially on MgO single crystal substrates with the same textures[17]. Using X-ray diffraction (XRD), Pt shows a strong (220) peak on (110) MgO substrate, a strong (111) peak on (111) MgO substrate, and a strong (200) peak and a weak (111) peak on (100) MgO substrate. The amorphous thermal oxide Si substrate is used as a reference with all three textures detected in the Pt films so that the Pt grain orientation is random. To determine the conventional ($\sigma_y$) and unconventional ($\sigma_x$ and $\sigma_z$) spin torque efficiencies, we used angular-dependent second-harmonic Hall (SHH) measurements and calculations (Supplementary Methods). A charge current was applied along the longitudinal direction of a Hall-bar structure, and then the first and second-harmonic Hall resistances ($R_{H,1\omega}$ and $R_{H,2\omega}$) were measured dependent on the angle φ of the in-plane applied magnetic field. In Fig. 1b, c, we show the experimental data and fitting of the SRT1/Mg/CFB stack grown on (100) MgO substrate with current $\mathbf{J}//[001]$. The effective spin torque efficiencies $\theta_{AD,x}$, $\theta_{AD,y}$, $\theta_{AD,z}$, and $\theta_{FL,y}$ are estimated to be (−0.083 ± 0.007), (0.102 ± 0.005), (−0.033 ± 0.010), and (0.0376 ± 0.009), respectively. Here, $\theta_{AD}$ and $\theta_{FL}$ refer to anti-damping (or damping-like) torque efficiency and field-like torque efficiency, respectively; and the subscript $x$, $y$, or $z$ denotes the spin polarization direction. Retaining the same stack structure, we changed the underlying substrates to (110), (111) MgO single crystals, and amorphous Si/SiO$_2$, and obtained the spin torque efficiencies of $x$ spin polarizations $\theta_{AD,x}$ (Fig. 1d) and of $y$ spin polarizations $\theta_{AD,y}$ (Fig. 1e) dependent on crystalline orientations and charge current directions. At $\mathbf{J}//[001]$, $\theta_{AD,x}$ reaches up to ~ −0.09, which is 4.5 times higher than the efficiency from $\mathbf{J}//[010]$, [011], [111] et al. This suggests a crystalline-texture-dependent anisotropy in spin torque efficiency. While $\theta_{AD,x}$ is maximized at $\mathbf{J}//[001]$, the conventional spin torque efficiency $\theta_{AD,y}$ is significantly reduced, implying spin polarizations reorientation occurring in this system. With a small resistivity ρ ~ 30 μΩcm, this work achieved the largest unconventional spin torque efficiency among the emerging SOC materials[15,18,19], and retained large conventional spin Hall conductivity $\sigma^y_{zx}$ of 3.4×10$^5$ $\hbar/2e$ $\Omega^{-1}$m$^{-1}$ which is competitive with commonly used heavy metals[20,21].

### Distinct dependence of the SOT properties on Co thickness

Our $[Pt(1)/Co(t)]_5$ SOC multilayers were magnetron-sputtered at room temperature on MgO single crystals or thermal oxide coated Si substrates. The numbers in parentheses represent the nominal thicknesses in nanometers, and $t$ denotes the thickness of Co. The in-plane magnetic layer CoFeB(2.5 nm) capped by MgO(1.5)/Ta(2) was deposited on top of the SOC multilayer for the SOT characterization and magnetization switching experiments. We used a thin insertion layer Mg(2) to smooth the surface and decouple the Co layers with the in-plane magnetized CoFeB layer while keeping a transparent interface for spin transport (Fig. 2a). All samples were as-deposited and measured at room temperature. In particular, we explored the two states of spin reorientation transition (SRT) with Co thickness of 0.159 nm and 0.507 nm, respectively[22]. Similar to the definition of SRT1, we define $[Pt(1)/Co(0.507)]_5$ as SRT2. The 2-D XRD spectra of the as-deposited stacks, including SRT1, SRT2, and Pt(4), all sputtered on the

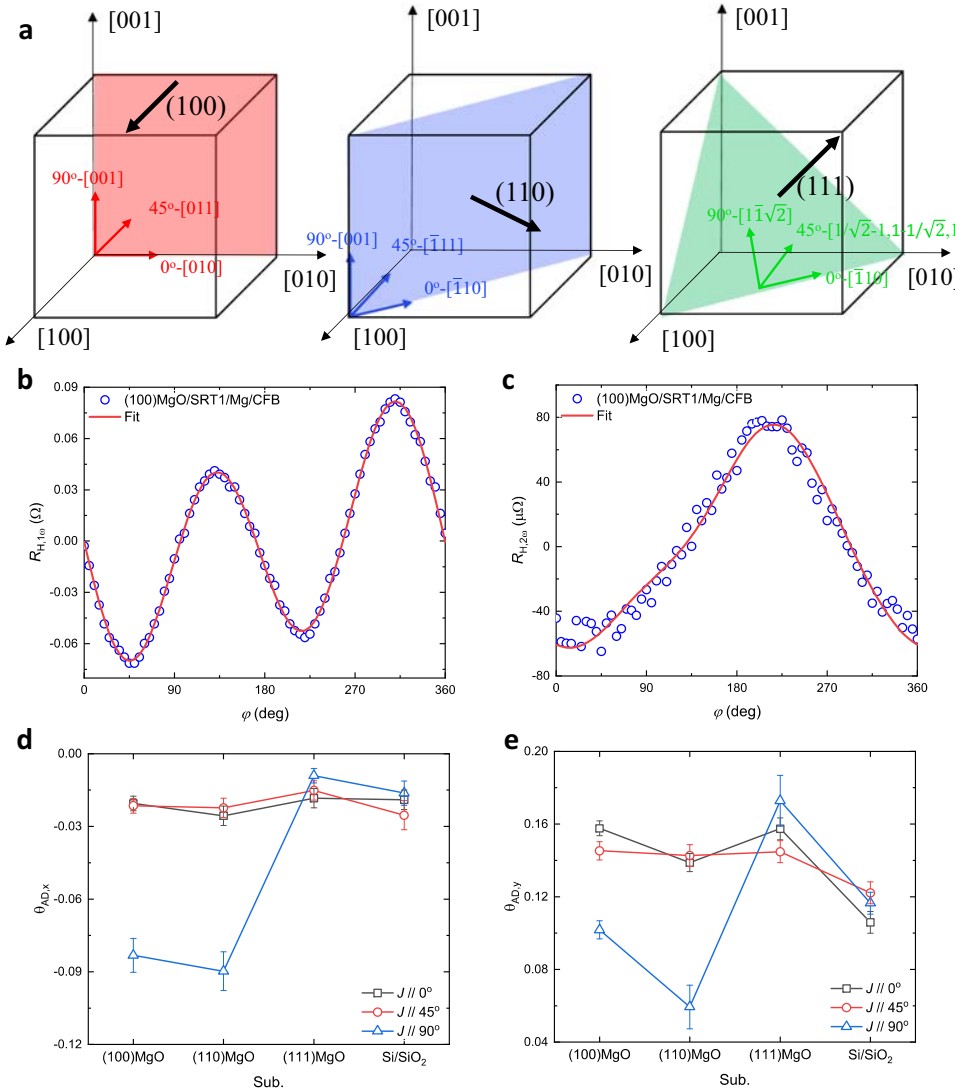

**Fig. 1 | SOT characterizations of the SRT1/Mg/CFB samples sputtered on crystalline or amorphous surfaces. a** The schematic of MgO crystallized substrates with plane orientated at (100), (110) and (111), as indicated by the three shadowed planes. The charge current **J** was applied at three local in-plane angles, 0°, 45° and 90°. This totally forms seven different directions in the crystalline coordinate system, [001], [011], [010], [$\bar{1}$11], [$\bar{1}$10], [1$\bar{1}\sqrt{2}$], and [1/$\sqrt{2}$-1,1-1/$\sqrt{2}$,1], as shown by the red, blue, and green arrows in the three types of MgO crystalline structure. **b, c** The $R_{H,1\omega}$ (**b**) and $R_{H,2\omega}$ (**c**) as a function of in-plane magnetic field rotation angle φ with field magnitude of 200 mT in stack SRT1/Mg/CFB grown on (100) MgO substrate with **J**//[001] direction. **d, e** Damping-like torque efficiencies of *x*-polarized spins (**d**) and *y*-polarized spins (**e**) of SRT1/Mg/CFB stack grown on (100), (110), and (111)-oriented MgO single crystals, and amorphous SiO₂ coated Si substrate (denoted as Si/SiO₂), and with various applied current directions. SRT1/Mg/CFB is the abbreviation of [Pt(1)/Co(0.159)]₅/Mg(2)/CoFeB(2.5) film stack with unit in nanometer. The error bars represent the parameters fitted with 95% confidence interval.

amorphous thermal oxide coated Si substrates (Fig. 2b), exhibit all three textures from Pt and no significant Co-thickness-dependent intensity. This implies weak texturing in the samples, with (111) preferred out-of-plane global orientation and random in-plane global orientation, which is related to the Pt cubic unit cell. One interesting discovery is an extra Bragg peak visible in the SRT2 sample occurring at 2θ = 34.6° with a weak signal, which is confirmed to come from the "−1" superlattice reflection consistent with the SRT2 stack thickness and indicated the formation of a Pt-Co superlattice structure in SRT2. The details of sample growth and film characterizations are presented in Methods.

Using micrometer-scale Hall-bar devices, we characterized magnetic properties through anomalous Hall effect (AHE) and planar Hall effect (PHE) measurements described in Methods, and SOT properties from angular-dependent SHH measurement. From Fig. 2c, d, CoFeB(2.5) is in-plane magnetized, SRT1 is close to a paramagnetic state, and SRT2 has a strong perpendicular magnetic anisotropy with

small AFM coupling among the multiple Co layers. The distortion of the sinusoidal first-harmonic Hall resistance $R_{H,1\omega}$ in SRT2-based stack (Fig. 2f), in contrast with SRT1-based stack (Fig. 2e), confirms the strong perpendicular magnetic component of SRT2. Surprisingly, the SOT characterizations ($R_{H,2\omega}$ signals in Fig. 2g, h) indicate that the unconventional spin torque efficiency $\theta_{AD,x}$ generated in SRT1 is significantly higher than that in SRT2. SRT1 (SRT2) stack shows negligible $R_{H,2\omega}$ signal compared to SRT1/Mg/CFB (SRT2/Mg/CFB) stack (Supplementary Fig. 2). Based on the experiment, the effective spin torque efficiencies $\theta_{AD,x}$, $\theta_{AD,y}$, $\theta_{AD,z}$, and $\theta_{FL,y}$ of Si/SiO₂/SRT1/Mg/CFB are estimated to be (−0.019 ± 0.004), (0.106 ± 0.006), (−0.0007 ± 0.0005), and (0.064 ± 0.006), respectively; and those of Si/SiO₂/SRT2/Mg/CFB are (−0.003 ± 0.002), (0.390 ± 0.013), (−0.089 ± 0.015), and (0.454 ± 0.010), respectively. The much higher $\theta_{AD,y}$ in SRT2 compared to SRT1 suggests that the conventional spin polarizations might be magnified by superlattice structure, strong interfacial energy, or perpendicular magnetization; while the

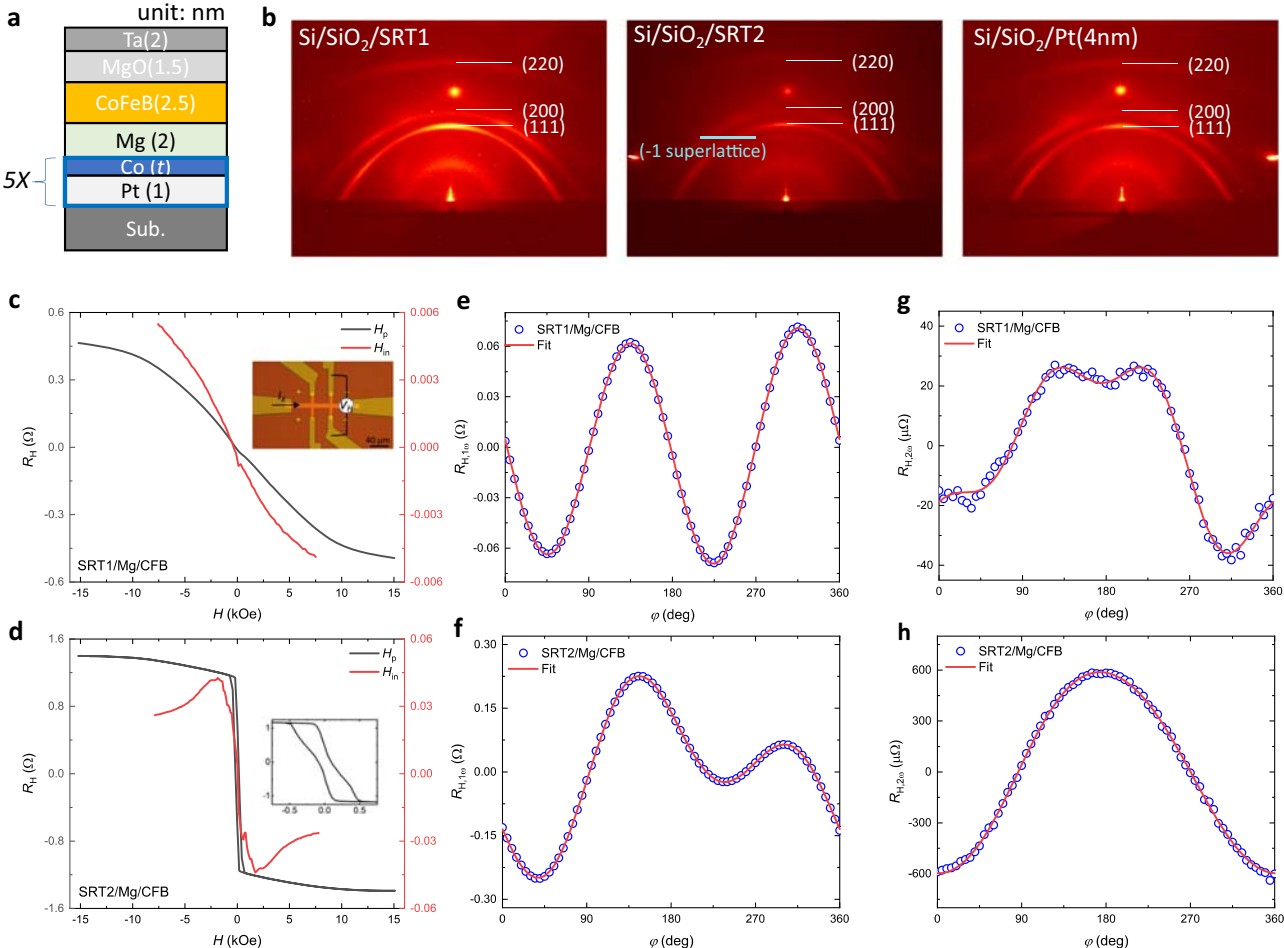

**Fig. 2 | The sample stacks and magnetic/SOT characterizations. a** The sample stack of sub./[Pt(1)/Co(t)]₅/Mg(2)/CoFeB(2.5)/MgO(1.5)/Ta(2) in the unit of nanometer. **b** 2-D XRD spectra of the as-deposited Si/SiO₂/SRT1, Si/SiO₂/SRT2, and Si/SiO₂/Pt(4 nm) films, showing all three crystalline textures (111), (200), and (220) in different SOC stacks grown on thermal oxide coated Si substrates. **c**, **d** Hall resistance $R_H$ *vs.* external magnetic fields in out-of-plane field $H_P$ and in-plane field $H_{in}$ in (**c**) SRT1/Mg/CFB and (**d**) SRT2/Mg/CFB. $R_H = V_H / I_x$. The inset figure in (**c**) is a Hall-bar structure. $I_x$ is the in-plane longitudinal current, $V_H$ is the transverse Hall voltage; the inset in (**d**) is a magnified view of the small $H_P$ field of sample SRT2/Mg/CFB. **e**, **f** the first-harmonic Hall resistance $R_{H,1\omega}$ as a function of the in-plane magnetic field rotation angle φ (200 mT) in (**e**) SRT1/Mg/CFB and (**f**) SRT2/Mg/CFB. **g**, **h** the second-harmonic Hall resistance $R_{H,2\omega}$ as a function of the in-plane magnetic field rotation angle φ (200 mT) in (**g**) SRT1/Mg/CFB and (**h**) SRT2/Mg/CFB. SRT1/Mg/CFB is the abbreviation of [Pt(1)/Co(0.159)]₅/Mg(2)/CoFeB(2.5) film stack, SRT2/Mg/CFB is the abbreviation of [Pt(1)/Co(0.507)]₅/Mg(2)/CoFeB(2.5) film stack with unit in nanometer.

unconventional spin torque efficiency $\theta_{AD,x}$ is greatly reduced by these factors, and could also be a result of competing Pt/Co interfacial effects and the Co bulk effects.

We also conducted two independent measurements for conventional SOT efficiency to confirm the accuracy of the angular-dependent SHH measurements presented above. One is the spin torque- ferromagnetic resonance (ST-FMR) measurement (Supplementary Methods). In the Si/SiO₂/SRT1/Mg/CFB sample, $\theta_{AD}$ is estimated to be (0.082 ± 0.013) (Supplementary Fig. 3a). For $\theta_{AD}$ calculation, the anti-damping torques from *x*-polarized spins $\tau_{AD,x}$ and from *y*-polarized spins $\tau_{AD,y}$ are mixed together because they are in the same direction along **m** × **z**[13]. Recalling the spin torque efficiencies of $\theta_{AD,x}$ ~ (−0.019 ± 0.004) and $\theta_{AD,y}$ ~ (0.106 ± 0.006) calculated from angular-dependent SHH measurement, the two methods are consistent. From ST-FMR measurement, we also obtained the magnetic damping constant α and the effective demagnetization $M_{eff}$ of (0.0088 ± 0.0003) and (0.944 ± 0.011) T, respectively. The other is the current-dependent SHH measurement (Supplementary Methods), which estimates the effective conventional spin torque efficiency $\theta_{AD,y}$ of Si/SiO₂/SRT1/Mg/CFB and Si/SiO₂/SRT2/Mg/CFB to be (0.095 ± 0.019) and (0.378 ± 0.023), respectively, in Supplementary Fig. 3b. These values

are in good agreement with the results from the angular-dependent SHH measurement within the sensitivity limit.

## Current-induced type-x SOT switching without external fields

In order to demonstrate the field-free in-plane magnetization switching of type-x SOT configuration assisted by unconventional *x*-polarized spins, we patterned the samples to micrometer-scale Hall-bar structures for differential planar Hall effect (DPHE) measurement[23] (Supplementary Methods). With a DC current *I* along the *x*-axis of the in-plane magnetization **M**, one can measure the Hall resistance $R_H$ in the Hall-bar structure. From the angular-dependent first-harmonic Hall resistance measurement in the SOT characterization as above, $R_H$ ~ sin2φ; therefore, a small bias field along the *y*-axis ($H_y$) makes opposite polarities of the gradient for +*x* and −*x* magnetizations, which helps to distinguish the magnetization directions in the Hall-bar device. The schematics of field-switching and current-switching measurements using DPHE method are shown in Supplementary Fig. 4a−c. The sweep field $H_x$ or write current pulse $I_W$ is applied for a duration of 1 ms. A reading current pulse $I_R$ is then applied and Hall voltage $V_H$ values are measured at a small $H_y$ bias of +3.2 Oe and −3.2 Oe. This small bias field is only applied during reading. The DPHE

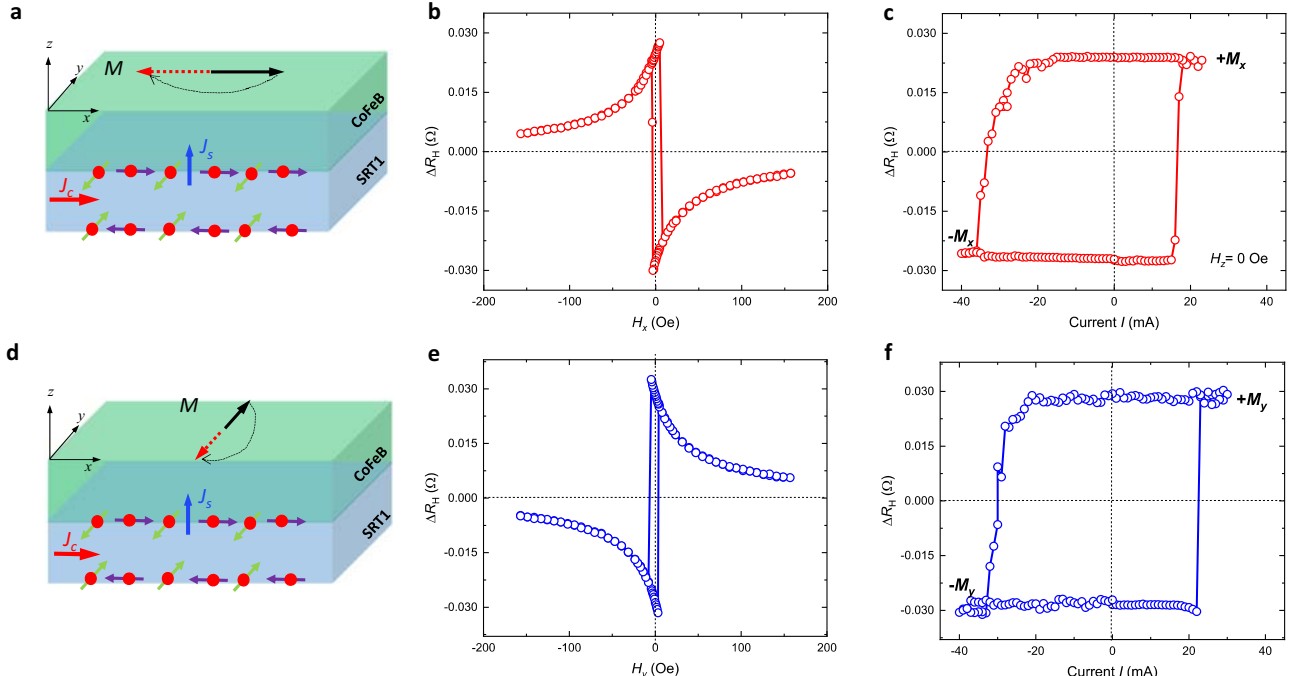

**Fig. 3 | Type-x and type-y SOT switching behaviors in the SRT1/Mg/CFB samples deposited on (100) MgO single crystal substrates. a** A schematic depiction of type-x SOT switching with spin polarizations of $\sigma_x$ and $\sigma_y$. The red spheres represent electrons, green (purple) arrows represent spin magnetic moment along $y$-direction ($x$-direction). $J_c$, $J_s$, and $M$ represent the charge current density, spin current density, and magnetization, respectively. **b, c** Field-switching and field-free current-switching in type-x geometry based on DPHE measurement with magnetic anisotropy along $x$ direction. **d** A schematic depiction of type-y SOT switching with spin polarizations of $\sigma_x$ and $\sigma_y$. **e, f** Field-switching and field-free current-switching in type-y geometry based on DPHE measurement with magnetic anisotropy along $y$ direction. SRT1/Mg/CFB is the abbreviation of the sample stack SRT1/Mg(2)/CoFeB(2.5)/MgO(1.5)/Ta(2).

signal is obtained as $\Delta R_H = (V_{H+} - V_{H-})/I_R$. On a reference sample stack Si/SiO$_2$/Pt(3)/Mg(2)/CoFeB(2.5)/MgO(1.5)/Ta(2), the type-x geometry can switch the magnetization at $\pm 30$ mA with external field $H_z = \pm 156$ Oe to break the inversion symmetry; current-switching was not observed in the absence of an external field using the same measurement approach, even when the write current range was increased until the device was damaged due to Joule heating (Supplementary Fig. 4d–e).

As the DPHE measurement confirmed the absence of unconventional spin polarizations in the reference sample, we then conducted the same measurement on SRT1/Mg(2)/CoFeB(2.5)/MgO(1.5)/Ta(2) stacks. Figure 3a schematically describes the type-x SOT switching assisted by in-plane unconventional spin polarizations $\sigma_x$ when a charge current density $J_c$ is injected along $x$-direction. Spin current density $J_s$ with spin polarizations of $\sigma_x$ and $\sigma_y$ flowing in $z$-direction both exert torques on magnetization $M$, and the torques from $\sigma_x$ break the inversion symmetry and switch magnetization deterministically in type-x geometry. Using DPHE method, we observed the field-switching of magnetization $M$ between $+x$ and $-x$ on a Hall-bar device with magnetic anisotropy along $x$ and an average coercivity ~5.1 Oe (Fig. 3b) on the (100)MgO/SRT1/Mg(2)/CoFeB(2.5)/MgO(1.5)/Ta(2) sample. Without external field $H_z$ to break the symmetry, a full switching of the in-plane magnetization has been observed at the current of $+16.6$ mA and $-33.5$ mA, with an average of 25.1 mA, or current density of 36.7 MA/cm$^2$ in the SOC layer. The asymmetry of switching current from $+x$ to $-x$ and $-x$ to $+x$ originates from the asymmetric coercivity due to the asymmetry from domain nucleation or edge pinning effect which resulted from sputtering and fabrication but could be reduced by post-annealing as needed. The much sharper switching behavior in the right branch also confirmed the asymmetry. Since a magnetic field along $y$ direction is not able to break the symmetry of the type-x SOT switching, the current-induced Oersted field along $y$ direction can also be excluded as the field-free switching mechanism. Similarly, we used

the Hall-bar device with magnetic anisotropy along $y$ axis to check the type-y switching using DPHE method as well. Figure 3d schematically describes the type-y SOT switching by in-plane conventional spin polarizations $\sigma_y$ when a charge current density $J_c$ is injected along $x$. Spin current density $J_s$ with spin polarizations of $\sigma_x$ and $\sigma_y$ flowing in $z$-direction both exert torques on magnetization $M$, and the torques from $\sigma_y$ break the inversion symmetry and switch magnetization deterministically in type-y geometry. Using DPHE method, we expectedly observed the field-switching between $+y$ and $-y$ on a Hall-bar device with the magnetic anisotropy along $y$ and an average coercivity ~5.0 Oe (Fig. 3e) on the same sample as of type-x SOT switching. Full switching of the in-plane magnetization along the $y$ axis has also been observed by DPHE method at the current of $+22.5$ mA and $-29.0$ mA, with an average of 25.8 mA, or current density of 37.7 MA/cm$^2$ in the SOC layer. Since the Hall-bar devices are in micrometer-scale, both type-x and type-y SOT switching are domain-wall mediated switching. Considering the close spin Hall efficiency values of $\sigma_x$ and $\sigma_y$ ($\theta_{AD,x} = -0.083$, $\theta_{AD,y} = 0.102$) and in-plane coercivity $H_c$ ($H_{c,x} = 5.1$ Oe, $H_{c,y} = 5.0$ Oe), the similar field-free switching current density in type-x ($J_{c,x} = 36.7$ MA/cm$^2$) and type-y ($J_{c,y} = 37.7$ MA/cm$^2$) geometries suggests a similar switching mechanism of spin polarization $\sigma_i$ on magnetization $M_i$ ($i = x, y$) in the regime of domain-wall switching. In addition, the opposite polarities of $\Delta R_H$ at positive (or negative) state of $M_x$ and $M_y$ also confirmed the opposite signs of spin Hall efficiencies $\theta_{AD,x}$ and $\theta_{AD,y}$.

Field-switching and current-switching results of SRT1/Mg(2)/CoFeB(2.5)/MgO(1.5)/Ta(2) samples grown on the Si/SiO$_2$ substrate also confirmed the field-free type-x SOT switching assisted by $x$-polarized spins (Supplementary Fig. 5). Even though $\theta_{AD,x}$ of the sample stacks sputtered on Si/SiO$_2$ substrates is 4.4 times lower than that of the samples on (100) MgO substrates, the switching current density is not 4.4 times higher as anticipated, but rather increased by only ~20% ($J_{c,x} = 43.7$ MA/cm$^2$). Therefore, in field-free type-x

SOT switching, $x$-polarized spins should not be viewed as the only source of SOTs that makes deterministic switching happen; more precisely, it acts as a disturbance to break the inversion symmetry and works in conjunction with $y$-polarized spins to achieve deterministic switching.

## Micromagnetic simulations

To understand the dynamics and mechanisms of $x$-polarized spins in assisting field-free type-x SOT switching, we numerically study magnetization dynamics with micromagnetic simulations, where the Landau-Lifshitz-Gilbert (LLG) equation is solved using MuMax3[24] (Supplementary Methods). In our simulation, we relax an elliptically shaped FM layer with a dimension of $128 \times 384 \times 2.5$ nm$^3$ and apply a charge current pulse with length of $t_{pulse}$. The switching current density $J_c$ is extracted by varying the current pulse length $t_{pulse}$ and the ratio of spin torque efficiencies $\theta_{AD,x}/\theta_{AD,y}$ while holding the magnitude of total polarized spins in Fig. 4. In Fig. 4a, we show the switching current density as a function of $t_{pulse}$, with different $\theta_{AD,x}/\theta_{AD,y}$ for type-x and type-y geometries. The type-x devices can be switched with sub-ns current pulses more efficiently without external magnetic fields, although the type-y devices require lower current density at longer current pulses. From Fig. 4b, the type-x devices show 2–3 times of benefits in regard to the switching current compared to the type-y devices, especially in the regime of short current pulses, which is promising to reduce the SOT switching current for SOT-MRAM applications and thus to scale down the SOT write transistors[25]. An experimental study about field-free SOT switching of the in-plane magnetized CoFeB nanodots[26] was conducted with various angles $\phi_{EA}$ between the magnetic easy axis and the current direction. This study verified the benefit of switching current density in field-free type-x geometry, especially with narrow current pulse. In type-x SOT switching, the increase of spin torque efficiency of $x$ polarizations will further reduce the switching current and broaden the range of current pulse width that can switch the MTJ compared to type-y SOT switching. In Fig. 4c, we present the switching trajectory of magnetization at $\theta_{AD,x}/\theta_{AD,y} = 1.0$ with shape anisotropy along $x$ direction under the current density of 252 MA/cm$^2$ with 1 ns pulse width. We note that the SOTs triggers the magnetization precession in both $x$ and $y$ directions; after initial precession for about 0.5 ns , the SOTs overcome the demagnetization and anisotropy fields and then align the magnetization along the $+x$ direction at ~0.8 ns, and sustains $+x$ direction even after the current turned off.

## Dependence of spin polarizations on Co magnetization

In [Pt/Co]$_N$ stacks, the Co thickness can modulate the magnetization through Co bulk effect and Pt/Co interface proximity effect[27], change the interfacial spin transport behavior through spin memory loss and spin backflow[28], and tune the spin/orbital currents through spin Hall effect (SHE) and orbital Hall effect (OHE) in Co and at the interfaces[29,30], etc. Therefore, we sputtered Co-wedged samples of [Pt(1)/Co($t_{Co}$)]$_5$ on Si/SiO$_2$ substrates with $t_{Co}$ from 0.159 nm to 0.395 nm, and investigated the thickness-dependent magnetic and SOT properties. Figures 5a, b are about AHE and PHE measurements, showing paramagnetic state at $t_{Co} = 0.159$ nm and in-plane ferromagnetic state

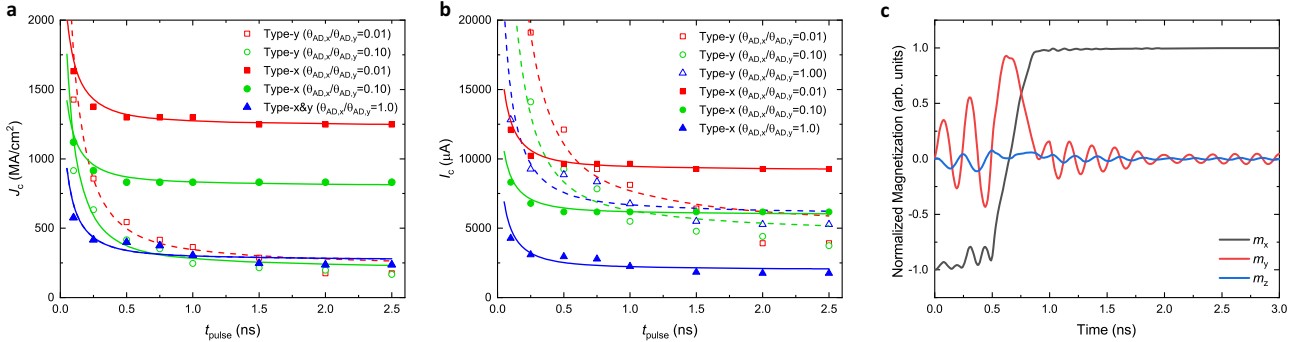

**Fig. 4 | Micromagnetic simulations of SOT switching behavior in the type-x and the type-y devices. a**, **b** The switching current density $J_c$ (MA/cm$^2$) (**a**), and the switching current $I_c$ (μA) (**b**), vs. current pulse width $t_{pulse}$ (ns) of field-free type-y and type-x SOT switching by both $x$ and $y$-polarized spins. The device is of ellipsoidal shape with SOC layer thickness at 5.8 nm. **c** The temporal profiles of $x, y,$ and $z$ components of the free layer magnetization $M$ with 1 ns switching current pulse at $\theta_{AD,x}/\theta_{AD,y} = 1.0$ in type-x geometry.

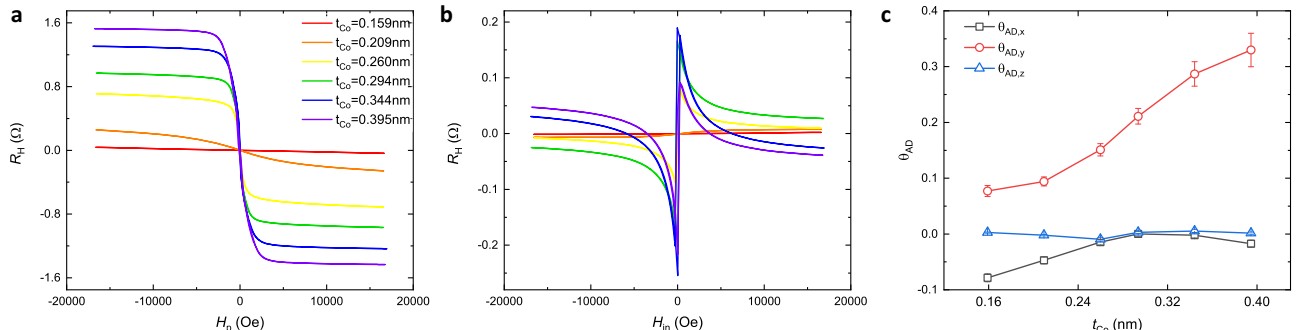

**Fig. 5 | Magnetic properties and SOT characterizations of sputtered Co-wedged samples on Si/SiO$_2$ substrates. a**, **b** The Hall resistance $R_H$ vs. perpendicular field $H_p$ (**a**) and in-plane field $H_{in}$ (**b**) at different Co thickness of 0.159 nm, 0.209 nm, 0.260 nm, 0.294 nm, 0.344 nm, and 0.395 nm in Co-wedged Si/SiO$_2$/[Pt(1)/Co($t_{Co}$)]$_5$/Mg(2) film stacks. **c** Anti-damping torque efficiencies of conventional spin polarizations ($\theta_{AD,y}$) and of unconventional spin polarizations ($\theta_{AD,x}$ and $\theta_{AD,z}$) at different Co thickness measured in Co-wedged Si/SiO$_2$/[Pt(1)/Co($t_{Co}$)]$_5$/Mg(2)/CoFeB(2.5) film stacks. The error bars represent the parameters fitted with 95% confidence interval.

at $t_{Co} = 0.209$ nm. The increasing of anomalous Hall resistance $R_A$ when Co gets thicker from 0.260 nm (~1 monolayer (ML)) indicates the increasing interfacial magnetic energy and perpendicular magnetic component in $[Pt(1)/Co(t_{Co})]_5$. The spin torque efficiencies (Fig. 5c) calculated from angular-dependent SHH measurement presented an increase of conventional spin torque efficiency $\theta_{AD,y}$ and a decrease of in-plane unconventional spin torque efficiency $\theta_{AD,x}$ as Co gets thicker. The out-of-plane unconventional spin torque efficiency $\theta_{AD,z}$ remains nearly 0 and is clearly not affected by $t_{Co}$.

### Strong orbital magnetic moment from low-dimensional Co as revealed by XMCD experiments

Inspired by the spin generation and transport in spintronics 2D materials[31], the appearance of non-vanishing unconventional spin polarizations $\sigma_x$ in ultrathin Co films (less than 1 ML) motivated us to explore the polarizations at the atomic level due to the low dimensionality in the $[Pt/Co]_N$ stacks. Liu et al. [32] reported a self-switching of perpendicular magnetization in CoPt single layer due to low crystal symmetry properties at the Co platelet/Pt interfaces. Gambardella et al. [33,34] reported giant magnetic anisotropy from the existence of both short- and long-range FM orders in low-dimensional Co sputtered on Pt substrates observed at $T = 10$ K. The orbital magnetic moment $m_L$ of Co sputtered on (111) Pt increased dramatically due to lowering dimension, from $m_L = 0.14$ $\mu_B$ per atom in bulk Co all the way up to $m_L = 0.79$ $\mu_B$ per atom in zero-dimension Co, where $\mu_B$ is the Bohr magneton. Due to the symmetry reduction at an atomically ordered surface with lower dimensions, the orbital magnetic moment of Co films and strong spin-orbit coupling induced by the Pt films together contribute to a stronger magnetic anisotropy of the Co atoms[35,36]. Accordingly, we performed XMCD measurements at room temperature[37] to probe the adatom magnetism by detecting X-ray absorption spectra (XAS) at the Co $L_2$ and $L_3$ edges (2p to 3d transitions) using left and right circularly polarized light in total electron yield mode. In Fig. 6a, the XAS signal is shown as a function of photon energy for left and right circularly polarized X-rays in samples SRT1 and SRT2. The XMCD signal is calculated as the difference of XAS signals in $\mu_+$ and $\mu_-$, and then normalized to the Co $L_2$ peak intensity (Fig. 6b). The larger XMCD signal at $L_3$ Co edge from the SRT1 sample than that from the SRT2 sample indicates a stronger orbital magnetic moment in SRT1. Determined by

$$\int_{L_3+L_2} (\mu_+ - \mu_-) dE = (C/2\mu_B) m_L. \tag{1}$$

We calculated that the orbital magnetic moment per atom in the SRT1 sample ($m_{L,SRT1}$) is 12.3% larger than that in the SRT2 sample ($m_{L,SRT2}$) at room temperature. Statistically, each Co layer in SRT1 is ~0.6 ML thick in a dimension of lower than 2D; while, each Co layer in SRT2 is ~2 ML thick in a dimension of 3D. We also investigated the Co-thickness-dependent XMCD signals and normalized to Co $L_2$ edge peak intensity. Figure 6c shows the XMCD signal of Co $L_3$ edge peak intensity dependent on $t_{Co}$ in the Co-wedged $[Pt(1)/Co(t_{Co})]_5$ stacks. The data reveals a trend of increasing Co $L_3$ peak intensity (in absolute value) and thereby stronger orbital magnetic moment with decreasing dimensionality of Co sputtered on Pt. This discovery is qualitatively consistent with the conclusions of [33].

### Discussion on the generation of unconventional spin polarizations

As revealed by the XMCD experiment, low-dimensional Co sputtered on Pt, e.g., SRT1, exhibits a strong orbital magnetic moment. Assuming that the spin-orbit coupling in the low-dimensional Co remains at the atomic level, the strong orbital magnetic moment should lead to spin-orbital scattering and thereby high spin-orbit torque efficiency. Since the heavy metal Pt with strong spin-orbit coupling does not break inversion symmetry, its bulk effect does not contribute to unconventional spin generation. The weakly in-plane magnetized Co next to Pt films is low-dimensional and does break inversion symmetry so that unconventional spin polarizations along the $x$-direction is allowed (Supplementary Fig. 6), and the significant $x$-polarized spin efficiency is highly likely to be originated from this.

The Co thickness dependent spin torque efficiencies $\theta_{AD,x}$ and $\theta_{AD,y}$ in SHH measurement (Fig. 5c) and orbital magnetic moment in XMCD measurement (Fig. 6c) can be alternatively explained by the magnetic spin anomalous Hall effect (MSAHE) and magnetic SHE (MSHE) in ferromagnets which generate intrinsic spin currents proposed by Amin et al. [29]. In this theory, the charge current applied along $x$ direction can be converted into spin current flowing in $z$ direction given by

$$\boldsymbol{Q}_z = \left[ (\sigma_{MSAHE} + \sigma_{MSHE}) m_y \hat{\boldsymbol{m}} + \sigma_{MSHE} \hat{\boldsymbol{m}} \times (\hat{\boldsymbol{m}} \times \hat{\boldsymbol{z}}) \right] E, \tag{2}$$

where $\sigma_{MSAHE}$ and $\sigma_{MSHE}$ refer to the spin Hall conductivities from MSAHE and MSHE, respectively. $E$ is the electric field from the applied charge current. The spin Hall conductivities computed in Co film with $\mathbf{E}//\mathbf{c}$ (crystalline-induced anisotropy along $c$-axis of the crystal structure) by first-principles calculations are $\sigma_{MSAHE} = -1004$ $\hbar/2e$ $\Omega^{-1}$m$^{-1}$ and $\sigma_{MSHE} = 1074$ $\hbar/2e$ $\Omega^{-1}$m$^{-1}$ [29]; therefore, $\boldsymbol{Q}_z$ is dominated by the term of $\sigma_{MSHE} \hat{\boldsymbol{m}} \times (\hat{\boldsymbol{m}} \times \hat{\boldsymbol{z}})$. According to the magnetization-direction dependent curves (Supplementary Fig. 7) from Eq. (2), the maximum spin current of $y$-polarized spins occurs when the magnetization is parallel with $zx$-plane. As Co gets thicker from 0.159 nm to 0.395 nm and magnetization tilts up from the $xy$-plane, the trend of thickness-dependent spin torque efficiency $\theta_{AD,y}$ follows that of the magnetization-dependent spin Hall conductivity $\sigma^y_{zx}$. Meanwhile, only Co magnetized in $xy$-plane can generate $x$-polarized spins. Therefore, we observe very small $x$-polarized spin current generated in $[Pt(1)/Co(t_{Co} > 0.260)]_5$ when a perpendicular magnetic component occurs;

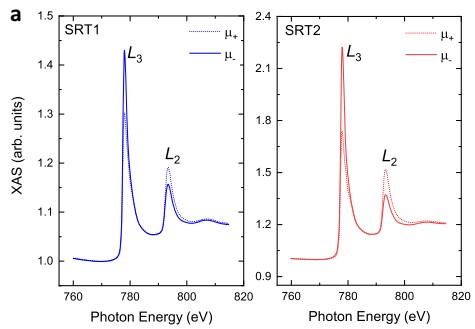
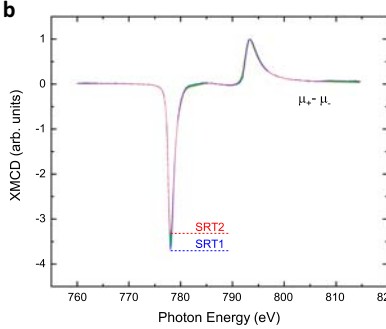
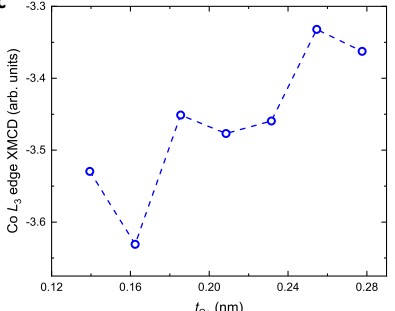

**Fig. 6 | Co XAS measurement and XMCD experiment on $[Pt/Co]_5$ multilayers on Si/SiO$_2$ substrates. a** The XAS signals of Co element in the SRT1 and SRT2 samples for parallel ($\mu_+$) and antiparallel ($\mu_-$) directions between light polarization and field-induced magnetization. **b** XMCD signals obtained by subtraction of the absorption spectra $\mu_+$–$\mu_-$ and normalized by the Co $L_2$ edge in the samples of SRT1 and SRT2. **c** Co-thickness-dependent XMCD signal of Co $L_3$ edge peak intensity in Co-wedged samples of $[Pt(1)/Co(t_{Co})]_5$ on Si/SiO$_2$ substrates.

and it is only in the multilayer with ultrathin Co films that we observe a significant and non-vanishing $x$-polarized spin current due to a weak magnetization in the $xy$-plane. This ultrathin Co multilayer in SRT1 is originally in a paramagnetic state, which is weakly magnetized in the $xy$-plane by the small magnetic fields from the stray field of the in-plane magnetized CoFeB and the field-like torque effective-field from the conventional spin polarizations.

In conclusion, we have demonstrated the field-free type-x SOT switching assisted by in-plane unconventional spin polarizations in [Pt/Co]$_N$ at room temperature. The generation of unconventional spin polarizations strongly depends on the crystalline texture of Pt as well as the low dimension of the sputtered Co. Independent measurements were conducted to calculate the spin torque efficiencies, with the spin torque efficiency of $x$ spin polarizations reaching up to -0.083. XMCD measurements confirmed the significance of the orbital magnetic moment and spin-orbital scattering in ultrathin Co and Pt/Co interfaces in generating in-plane unconventional spin polarizations by MSHE. Based on the micromagnetic simulation, the performance of type-x SOT switching in the presence of $x$-polarized spins excels over other configurations with regard to switching energy and latency, especially at sub-ns current pulse, which is promising for high-speed magnetic memories. This work provides an important milestone in the field of type-x SOT switching with CMOS-compatible materials and deposition methods for all-electrical manipulation of magnetic materials in the pursuit of high-speed, high-density, and low-energy non-volatile memory.

## Methods

### Sample deposition and thin film characterization
The [Pt/Co]$_N$ multilayer thin films for the magnetic and SOT characterizations, X-ray related measurement, and magnetization switching measurement were deposited on either Si/SiO$_2$ substrates or crystallized MgO substrates at room temperature by sputtering Pt (99.99% pure) target and Co (99.99% pure) target layer by layer in AJA magnetron sputtering system with a base pressure of $2.0 \times 10^{-8}$ Torr or lower. A 200 Oe-magnetic field generated by the N-S magnets attached to the wafer holder was applied during the deposition of all films. Targets Pt, Co, Mg, Co$_{20}$Fe$_{60}$B$_{20}$, and Ta were dc sputtered at 25 W, 50 W, 15 W, 25 W, and 25 W power at 2 mTorr Argon atmosphere, with deposition rates at 0.300, 0.183, 0.205, 0.057, and 0.154 Å/s, respectively. The MgO layer was rf sputtered at 100 W power and 0.6 mTorr Argon atmosphere, with a deposition rate of 0.025 Å/s.

The 2-D X-ray diffraction (XRD) spectra of [Pt/Co]$_N$ multilayers sputtered on Si/SiO$_2$ substrates were measured at Stanford Nano Shared Facilities (SNSF) using Bruker D8 Venture single crystal diffractometer with Cu Ka (8.04 keV) radiation in grazing incidence geometry. The X-ray absorption spectra (XAS) were conducted by detecting the fluorescence yield (FY) from the direct excitation of photons from the absorbing material by sweeping the in-plane magnetic fields or the photon energy at room temperature (300 K) at Advanced Light Source (ALS) beamline 4.0.2. The X-ray beam is focused on a waveguide attached to the sample and the FY signal is detected by a photodiode. The elliptically polarizing undulator (EPU) was set at +0.9 or −0.9.

### Device fabrication and electrical characterization
The multilayer thin films were patterned into Hall-bar structures for the magnetic characterization, Hall measurement, and magnetization switching experiments using a photolithography process with positive photoresist SPR3612 exposed by Heidelberg MLA 150 at Stanford Nanofabrication Facility (SNF). The width of the Hall-bar is designed at 2 μm, 5 μm, 10 μm, and 20 μm, and the length of it is 110 μm. The separation between the centers of Hall-crosses in Hall-bar device is 40 μm. The direction of the current channel in the Hall-bar is designed with different orientations of 0°, 45°, 90°, and 135° with respect to the

horizontal direction of the sample. The horizontal direction is defined as the direction of induced magnetic anisotropy from the magnets during sputtering for films on Si/SiO$_2$ substrates, or is defined along one edge of the crystallized MgO substrates. For the ST-FMR measurement, thin films were patterned into rectangular strips with dimensions of 20 μm by 30 μm. The strip is designed orientated 45° from the horizontal direction of the sample to achieve the highest signal from spin precession. The photolithography process of the strips is similar to that of Hall-bar structures. Followed by the photolithography process, the Argon ion-beam etching by Intlvac Ion Beam Mill Etcher at SNSF was used to shape the device geometry and to etch down to the insulating layer of the substrates. The Ti (10 nm)/Au (150 nm) metal layer was deposited after the second step of photolithography for electrical contact pads using Kurt J. Lesker E-beam Evaporator at SNSF. A lab-built measurement system controlled by Labview program with GPIB communication was used to measure the electrical properties for magnetic and SOT characterizations. Two Lock-in amplifiers SR830 and one Keithley 6221 current source were used to probe first and second harmonic voltages and to supply current, respectively. A set of magnet coils actuated by Kepco Power Supply supplied magnetic fields of up to 1.7 T. An Agilent HP 83624B was used to supply the rf current for the ST-FMR measurement with the power of 20 dBm and frequency of 2–20 GHz. The dc voltage was measured with a Keithley 2000 multi-meter.

## Data availability
The data that support the findings of this study are available from the corresponding authors upon reasonable request.

## Code availability
The custom code that supports the findings of this study is available from the corresponding authors upon reasonable request.

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

## Acknowledgements

This research was supported in part by TSMC University Joint Development Program (JDP) under award SPO135237. Part of this work was performed at the Stanford Nano Shared Facilities (SNSF)/Stanford Nanofabrication Facility (SNF), supported by the National Science Foundation under award ECCS- 2026822. This research used resources of the Advanced Light Source, a U.S. DOE Office of Science User Facility under contract No. DE-AC02-05CH11231. Certain commercial equipment and instruments are identified in this paper to foster understanding. Any mention of commercial products is for information only; it does not imply recommendation or endorsement. The authors would also like to acknowledge Dr. Chong Bi, Dr. Peng Li and Dr. Carlos H. Diaz for fruitful discussions.

## Author contributions

F.X. conceived, designed, and coordinated the research with contributions from S-J.L., M.S., C-M.L., W.T., and S.X.W. S.X.W. supervised the study. F.X. deposited thin films, performed XRD measurement, fabricated Hall-bar devices and ST-FMR devices, carried out ST-FMR, SHH, and switching measurements with contributions from E.T., M.S., C-M.L., C.K., P.S., A.V., Y-L.H. W.H., E.T., and Y-L.H performed micromagnetic simulations. F.X. performed data analysis and wrote the manuscript with contributions from E.T., W.H., W.T., X.B., and S.X.W. All authors discussed the results and commented on the manuscript.

## Competing interests

The authors declare no competing interests.
