## [Peer Review File · Nature Communications]

Reviewers' Comments:

Reviewer #1:

Remarks to the Author:

The work by Xue et al. reports the unconventional spin polarization generated by [Pt/Co]_n multilayers. The authors comprehensively quantify the spin torque efficiencies associated with different spin polarization and demonstrate the field-free type-x switching. Micro-magnetic simulations are also performed to discuss the features of x-polarization driven spin torque switching. XMCD measurements reveal larger orbital magnetic moment in SRT1 sample with lower dimension. These results provide strategy to generate x polarized spin and highlight its superior advantages in spin torque application.

However, the manuscript has a chaotic organization which makes it is extremely difficult to read and understand. The abstract length, the number of sample and number, the measurement technique are excessive. On the other hand, the quantification and derivation procedures in SHH, ST-FMR and DPHE are too simplified to produce convincing results. Overall, this work is not suitable to publish in its present form unless the authors could greatly improve the organization and make more concise and convincing presentation. Below please see my detailed comments:

1. Below statements should be elaborated more clearly and provided with appropriate references:

a. Line 84: "...type-x SOT switching scheme has the advantages of being faster than the classic type-z switching." The authors should specify the magnetization anisotropy in the discussion. For the scenarios of (z-polarized spin, M_z) and (x-polarized spin, M_x), the spin polarization and M are collinear and such that the speed and efficiency should be similar?

b. Line 91: "Therefore, the field-free type-x SOT switching is much easier than type-z in view of the unconventional spin efficiency required in the deterministic switching." There is no evidence to support the smaller spin efficiency required in the deterministic switching for type-x as compared to type-z. In Lee et al., Sci. Rep. 10, 1772 (2020) and Fukami et al., Nat. Nanotechnol. 11, 621 (2016), the critical switching current density of type z is smaller than type-y at a factor of damping constant. However, there is no such direct comparison between type x and type z/type y.

c. No basis and references for Equations S2 ~ S9. What are the AD and FL effective field directions for each spin polarization? Have the authors considered the interplay (or coupling) between AD and FL effective fields through PHE or AHE? What are the detailed fitting procedures and how large is the fitting uncertainty since the fitting involve ~10 parameters?

d. Line 162, how to unambiguously identify the small AFM coupling in the sample? Does the AFM coupling play a role?

e. Line 168 ~ 170, why not provide theta_{FL,x} and theta_{FL,z}?

f. In the ST-FMR section of methods, the model does not consider other spin polarization so the derived spin torque efficiency cannot make direct comparison to the SHH results.

g. Line 362, there is no indicative trend of "close to saturation".

h. Line 373, how to understand "The ultrathin Co multilayer in a paramagnetic state is likely to be at the ideal magnetization orientation close to 45deg in the xy-plane." The paramagnetism and ferromagnetism co-exist?

i. Line 261, how to understand "Spin currents J_{ox} and J_{oy} both flow in z-direction and exert torques on magnetization M with σ_y breaking the inversion symmetry."

2. The schematic of stacking all three MgO vertically in Fig. 1a is inappropriate and misleading. And for a better logic flow, Fig. 1 d,e should be moved to the position before Fig. 1 b,c.

3. SRT in Fig. 1d,e and pure Pt sample in line 213 are not defined beforehand.

4. Table 1 makes little meaning as the prior results are too few. It may provide wrong indication of large unconventional spin polarization in this work.

5. The coercivity of the SRT1 and SRT2 sample are only few Oe which make the switching experiment very suspicious. The switching could be related to the current induced Oersted field or the STT effect between [Pt/Co]_n and CoFeB etc.

6. At the end, the authors should try to make more concise presentation and discussion. One will be confused by the MgO orientation dependence, the Co thickness dependence, the SAHE/SHE/MSHE/MOHE mechanisms etc. They are very much out of organization.

Reviewer #2:

Remarks to the Author:

Xue and colleagues report investigations of current-induced switching of type-x SOT devices with [Pt/Co]N underlayers. Type-X SOT devices attract increasing attention owing to high-speed switching. Compared to type-y devices, the channel width of type-x can be reduced, leading to a relatively small switching current. However, an out-of-plane external field is required to induce deterministic switching in conventional structures. The results of the harmonic Hall measurement show the unconventional spin current, which is corresponding to the x-component of spin polarization. Using this unconventional spin current, authors achieve field-free SOT switching in type-x devices. The XMCD measurement indicates magnetic anisotropy of Co layer which provides a possible mechanism of the unconventional spin current. This work is a high-quality experimental study and may be of interest to a broad community but I encourage the authors to address the following comments before the publication.

1. In Fig.1(b), the current direction dependence of spin torque efficiency due to x-spin polarization ($\theta_{AD,x}$) is displayed and large efficiency is observed in stacks on MgO(100) and (110) when $J // 90\text{deg}$. However, for stacks on MgO(100) substrate, $J // 0\text{ deg}$ and $J // 90\text{ deg}$ indicate the equivalent experimental configuration and similar spin torque efficiency is expected. I suggest that the authors discuss this point.
2. There is a paper reporting the SOT-induced switching of in-plane magnetized CoFeB nanodots with various angles (ϕ_{EA}) between the magnetic easy axis and current direction [Takahashi et al., Appl. Phys. Lett. 114, 012410 (2019)]. The previous paper demonstrates field-free SOT switching above ϕ_{EA} of 10 deg by using pulse current of 1.7 ns – 100 ms. I understand that the mechanisms of SOT are different between the submitted manuscript and the previous study, however, the switching property such as threshold current vs pulse width might not be largely different. I propose that the authors compare this previous study with the submitted manuscript and discuss the difference in switching properties.
3. In figure 2, out-of-plane RH-H curves and angle dependence of harmonic Hall resistance are displayed. Because Co/Pt multilayer of SRT1 sample is close to a paramagnetic state, the anomalous Hall resistance of CoFeB layer is dominant in Fig.2(c), and the result of the 2nd harmonic Hall resistance (Fig.2g) reflects the magnitude of SOT acting on CoFeB. Meanwhile, in SRT2 sample, the Hall resistance is mainly due to Co/Pt multilayer, therefore, the result of Fig.2h may be originating from the SOT on Co/Pt layer. Before the comparison of SOT efficiency as shown in Fig.7c, I would suggest that the authors extract the magnitude of SOT acting on CoFeB layer even for SRT2 sample.
4. On page 19, it is mentioned that the magnetization orientation is close to 45 deg away from the film normal direction. I think that the 45 deg tilt of magnetization is not very convincing because it requires another different magnetic anisotropy rather than both perpendicular and in-plane uniaxial magnetic anisotropies. Experimental evidence for such unconventional magnetic anisotropy is needed. Otherwise, the discussion part of the submitted manuscript needs to be toned down.
5. In line 386 of page 20, ref 34 is cited as a report on the self-switching of perpendicular magnetized CoPt. But ref 34 reports the manipulation of exchange bias by SOT in Pt/Co/IrMn. The citation part should be corrected.

Reviewer #3:

Xue et.al. reported the field-free switching of type x in-plane magnetization by pure electric current which is paramount important for SOT-MRAM application with fast switching since type x in-plane shows more than 10 times faster magnetization switching induced by electric current than those of type y in-plane and type z perpendicular magnetization. Particularly, the materials structure used in the manuscript is compatible with industry fabrication process. Fundamentally, they found an unconventional spin current with spin polarization in x direction in a simple material-ultrathin Co layer in Co/Pt multilayer, which is demonstrated by the detailed electric transport and material characterization. Spin current with x spin polarization is attributed to the strong spin orbital coupling of ultra-thin Co layer, which has never been observed before. The presented results are both fundamentally and technically interesting, especially important in the fields of spintronics and condensed matter physics. Therefore I would recommend it for publication in Nature Communication after they address the following technical comments.

- 1) The figure 1 schematically shows relationship between crystal direction and current direction. It is confused that the experiment seems not consistent with this drawing. e.g. how electric current // [001] direction is applied? It is quite usual that one use [001] direction to refer to film normal. In Fig.1 b, $J//90^\circ$ for the films grown on Si/SiO₂, MgO (111) are different from MgO(100), MgO(110), it is easy to cause the misleading. Same for $J//45^\circ$. In addition, Co layer grown on MgO single crystal substrate should be fcc structure, therefore, [001] and [010] should be equivalent, why the AD_x and AD_y are different for the sample on MgO(100) substrate?
- 2) Do the samples SRT1 and SRT2 any shape anisotropy? In Fig.2c, which is the direction of applied field, along current direction or transverse to current direction? since it is measured for the whole stack, CFB in figure 2c is also superparamagnetic?
- 3) “The asymmetry of switching current from +x to -x and -x to +x originates from the asymmetric coercivity due to unbalanced energy states of domains in +x and -x without annealing.” Why there exists unbalanced energy state? What means “without annealing”? do you mean that after annealing, there will be unbalanced energy state?
- 4) I noticed that the authors stated “It should be noted that the small bias field H_y is for the purpose of DPHE measurement alone and is not able to break the symmetry of the type-x SOT switching.” Do they apply a bias field during switching field? If yes, authors may try during switching, do not apply the bias field. Only applying the bias field to distinguish the magnetization direction after switching.
- 5) In Fig.7, the authors used wedge samples which will introduce the asymmetry in the wedge direction, which may induce the different spin polarization. It would be good not use the wedge sample to evaluate the thickness dependence on the different spin polarization induced AD torque.

Reviewer #1 (Remarks to the Author):

The work by Xue et al. reports the unconventional spin polarization generated by $[\text{Pt}/\text{Co}]_n$ multilayers. The authors comprehensively quantify the spin torque efficiencies associated with different spin polarization and demonstrate the field-free type-x switching. Micro-magnetic simulations are also performed to discuss the features of x-polarization driven spin torque switching. XMCD measurements reveal larger orbital magnetic moment in SRT1 sample with lower dimension. These results provide strategy to generate x polarized spin and highlight its superior advantages in spin torque application.

However, the manuscript has a chaotic organization which makes it is extremely difficult to read and understand. The abstract length, the number of sample and number, the measurement technique are excessive. On the other hand, the quantification and derivation procedures in SHH, ST-FMR and DPHE are too simplified to produce convincing results. Overall, this work is not suitable to publish in its present form unless the authors could greatly improve the organization and make more concise and convincing presentation. Below please see my detailed comments:

Thanks much for your insightful comments on improving the organization and presentation of our study and readability of this manuscript. *We have carefully considered the comments and addressed the questions point by point to the best of our ability.* Revisions are highlighted in the manuscript.

1. The abstract has been refined to make it clearer and shorter.
2. We think the abundance of the thin film samples and measurement techniques contained in the manuscript is a strength, albeit confusing at times, but we have clarified their purpose and necessity when we first introduce them in the main manuscript. We sincerely hope that the logic flow and presentation in the revised manuscript are now much better after we incorporate your input.
3. SHH, ST-FMR, and DPHE are quite commonly used measurement techniques in characterizing spin efficiencies and probing magnetic switching which have been thoroughly described in the references. Now we have added clearer derivation process of the theory and equations for data processing in the Supplementary Information. We have also revised measurement methods in Supplementary Information for better readability.

4. We are sorry that the 1st round submitted manuscript is not easy to read and understand. As you have suggested, we have added clear subheadings and some transitional sentences throughout the revised manuscript to navigate the readers, we have also inserted an introductory paragraph before RESULTS section (**Lines 86-91** in the revised manuscript).to summarize the order of the work presented in the manuscript, and also trimmed down the explanation/discussion section to make it easier to be understood.

1. Below statements should be elaborated more clearly and provided with appropriate references:

- a. Line 84: "...type-x SOT switching scheme has the advantages of being faster than the classic type-z switching." The authors should specify the magnetization anisotropy in the discussion. For the scenarios of (z-polarized spin, M_z) and (x-polarized spin, M_x), the spin polarization and M are collinear and such that the speed and efficiency should be similar?

Following the definitions from Fukami et al., Nat. Nano. 2016 [4], type -x, -y, and -z refer to the switching layer with magnetic easy axis along the current (x) direction, y direction, and the out-of-plane (z) direction, respectively. This does not refer to any spin polarization direction. Clarifications have also been made in the Introduction section (**Lines 43-45** in the revised manuscript).

The sentence in Line 84: as clarified above, type-x refers to magnetic anisotropy along x direction, and type-z along out-of-plane, regardless of the directions of the spin polarizations in the system. Here, we can explain type-x switching being faster than type-z switching from macro-spin dynamics or from micromagnetic simulation. The original sentence has also been rephrased to clarify the reasoning (**Lines 69-71** in the revised manuscript).

- 1) In the book of "Physics of Ferromagnetism" by Sōshin Chikazumi, the discussion of spin dynamics indicates that the out-of-plane demagnetization in in-plane switching, e.g., type-x switching, helps such a switching action. In other words, the type-x configuration has an in-plane shape anisotropy when M rotates out of plane on top of the x-axis anisotropy which assumes M staying in plane. The composite anisotropy makes type-x anisotropy "stiffer" and thus magnetization precession trajectory distorted and faster. While in type-z switching, since easy axis is along z, there is no such an additional shape anisotropy to help

align M to z -axis, and thus the precession trajectory is undistorted and takes longer to complete one precession cycle.

- 2) Reference [4] of the manuscript implies qualitatively similar dynamics between type- x and type- z switching. However, considering the huge differences in magnetic anisotropy and device geometry, a more nuanced approach is necessary to compare type- x and type- z switching in terms of switching current density and speed. A relatively fair way is to compare type- x and type- z switching under the same thermal stability ratio Δ . As shown by our micromagnetic simulation in Fig. r1-1, in type- x switching, three different scenarios are simulated: field-assist conventional SOT switching (a), field-free x -spin assisted SOT switching (b), and x -spin-only SOT switching (c); similar scenarios are shown for type- z switching in (d)-(f). In type- x , a small x -polarized spin torque efficiency ($\theta_{AD,x}/\theta_{AD,y}=0.065$ in Fig. r1-1b) is able to achieve field-free type- x switching in 0.5 ns with similar current density and dynamics as in Fig. r1-1a. In comparison, a relatively big z -polarized spin torque efficiency ($\theta_{AD,z}/\theta_{AD,y}=0.268$ in Fig. r1-1e) is needed to achieve field-free type- z switching in 1.5 ns with similar current density and dynamics as in Fig. r1-1d. For scenarios of x -polarized-spin-only type- x SOT switching (Fig. r1-1c) and z -polarized-spin-only type- z SOT switching (Fig. r1-1f), they are respectively close to type- x STT and type- z STT switching. As shown in the plots, their dynamics are similar and type- x (~ 0.2 ns) is still significantly faster than type- z (~ 0.5 ns). Therefore, their spin dynamics are not the same and are consistent with the macro-spin dynamics considerations in Chikazumi.

Fig. r1-1 Dynamic magnetization trajectories of type-x switching (a)-(c), and type-z switching (d)-(f). Current density, external field, and spin polarizations have been indicated in the plots.

b. Line 91: “Therefore, the field-free type-x SOT switching is much easier than type-z in view of the unconventional spin efficiency required in the deterministic switching.” There is no evidence to support the smaller spin efficiency required in the deterministic switching for type-x as compared to type-z. In Lee et al., Sci. Rep. 10, 1772 (2020) and Fukami et al., Nat. Nanotechnol. 11, 621 (2016), the critical switching current density of type z is smaller than type-y at a factor of damping constant. However, there is no such direct comparison between type x and type z/type y.

We want to first clarify that the switching current density of type-z is higher than that in type-y from the Fukami et al., Nat. Nano. 2016 reference [4]. If comparing the two J_c equations of type-z and type-y as put below, the factor of damping constant in type-y will reduce the absolute number of J_c due to the damping constant being less than 1 in general (e.g., CoFeB is 0.007 to 0.1). In addition, the magnetic anisotropy H_k is generally much smaller in type-x than in type-z, as shown in the table below [4].

$$J_C = \frac{2e M_S t_F}{\hbar} \frac{1}{\theta_{SH}^{eff}} \left(\frac{H_K^{eff}}{2} - \frac{H_x}{\sqrt{2}} \right) \quad J_C = \frac{2e \alpha M_S t_F}{\hbar} \frac{1}{\theta_{SH}^{eff}} \left(H_{K,in}^{eff} + \frac{H_{K,out}^{eff}}{2} \right)$$

Table 1 | Comparison of structures of types x, y and z.

	Type z	Type y	Type x
$\mu_0 H_K^{\text{eff}}$ (mT)	260 ± 10	4.2 ± 0.5	4.04 ± 0.01
$E/k_B T$	42 ± 2	46 ± 4	45.5 ± 0.2
J_C^0 (10^{11} A m $^{-2}$)	20 ± 1	1.0	4.3 ± 0.2
c_j^{th} (mT)	130	1.58	19.95
$\theta_{\text{SH}}^{\text{eff}}$	-0.25 ± 0.02	-0.08	-0.22 ± 0.01

Evaluated effective anisotropy field H_K^{eff} , thermal stability factor $E/k_B T$, critical current density at zero magnetic field J_C^0 , threshold value of spin-transfer torque coefficient c_j^{th} at zero field calculated from macrospin simulation, and corresponding effective spin Hall angle $\theta_{\text{SH}}^{\text{eff}}$ for devices with type z, y and x geometries fabricated in this work.

Referring to the micromagnetic simulation study in Lin et al., 2021 IRPS [2], “To enable a field-free deterministic switching, the $\sin\theta$ ($\sin\phi$) in type-x (z) has to be larger than 0.017 (0.34) as shown in Fig. 6c” (see Fig. r1-2). This confirmed our statement that the unconventional spin torque efficiency required in x-polarized-spin assisted type-x switching is smaller than that in the z-polarized-spin assisted type-z switching. The data was also referred and rephrased in Introduction section (**Lines 73-76** in the revised manuscript). Our micromagnetic simulation (Fig. r1-1) for question 1(a) also confirmed this conclusion.

Fig. r1-2 Micromagnetic simulation: switching current vs. switching time for field-free type-x, type-y, and type-z switching [2]

c. No basis and references for Equations S2 ~ S9. What are the AD and FL effective field directions for each spin polarization? Have the authors considered the interplay (or coupling) between AD and FL effective fields through PHE or AHE? What are the detailed fitting procedures and how large is the fitting uncertainty since the fitting involve ~10 parameters?

The detailed derivations of Supplementary Eqs. (1)-(9) are referred from the Supplementary references [i-iii], and the direct equations are from the Supplementary Information in DC et al., Nat. Mat. 2023, Apr. 3 [15]. We have clarified the references in **Line 43** in the revised Supplementary Information.

The effective field H_{FL} is along $\mathbf{m} \times (\mathbf{m} \times \boldsymbol{\sigma})$ direction and the effective field H_{AD} is along the $\mathbf{m} \times (\mathbf{m} \times (\boldsymbol{\sigma} \times \mathbf{m}))$ direction. Here, $\boldsymbol{\sigma}$ is spin polarization and \mathbf{m} is magnetization. $H_{FL,\sigma}$, $H_{AD,\sigma}$ ($\sigma=x, y, z$), and \mathbf{m} are always orthogonal with each other; therefore, there is no coupling between H_{FL} and H_{AD} .

The second harmonic Hall resistance is derived from the first harmonic Hall resistance which has considered the coupling between R_{AHE} and R_{PHE} . As the paragraph indicated, the parameters are extracted from either field-dependent or angular-dependent first/second harmonic Hall resistances. We have elaborated the fitting procedures in **Lines 57-64** in the revised Supplementary Information.

As for the uncertainty, the nonlinear fitting algorithm gives the error bar for 95% confidence intervals. All the fitted numbers, including spin torque efficiency values, from different measurement methods, are presented with error bars in this manuscript. For example, $\theta_{AD,x} = -0.083 \pm 0.007$, has an uncertainty of less than 10%; while others may be lower or higher in the range from 2% to 30%. The spin torque efficiencies we calculated are all presented with error bars (Fig. 7c and citations in **Lines 352 and 360** in the revised manuscript).

d. Line 162, how to unambiguously identify the small AFM coupling in the sample? Does the AFM coupling play a role?

From the inset R-H loop of Fig. 2d in the manuscript, we can decide whether it is of AFM coupling. Qualitatively, the loop becomes narrower when near 0 field. Using a simplified two-FM in-plane system to represent the $[Pt/Co]_5$, we plotted a schematic in Fig. r1-3 to explain the small AFM in the sample. When external field is 0, the total magnetization in in-plane direction is less than half of saturation value. If there is no AFM coupling in this stack, the in-plane magnetization should be close to 1 near 0 field, following the trend of the green dash lines.

Fig. r1-3 R-H loop of SRT2/Mg/CFB. Arrows represent the magnetization.

There are some research reporting spin currents generated by AFM materials, such as Cr_2O_3 in Li et al., Nature 2020, or assisting field-free switching due to exchange bias in Lau et al., Nature Nanotechnology 2016. However, we don't think our structure provides significant unconventional spin torque efficiency from AFM coupling: 1) The $[\text{Pt}/\text{Co}]_N$ stack with small AFM, notated as SRT2 in the manuscript, didn't show large unconventional spin torque efficiency as in the stack of SRT1 which has no AFM coupling. 2) The SHH measurement is conducted under a large in-plane external field up to ~ 7 kOe which saturated the magnetization along the uniaxial direction, but unconventional spin torque efficiency is still there when the AFM coupling state doesn't exist.

e. Line 168 ~ 170, why not provide $\theta_{\text{FL},x}$ and $\theta_{\text{FL},z}$?

First, field-like spin torques are not our focus because the dominant current-induced SOT switching mechanism for all the configurations is still damping-like torque rather than field-like torque, even though some researchers report field-like-torque assisted/suppressed SOT switching recently, e.g., Zhuo et al., Science China Physics, Mechanics & Astronomy 2022. The materials we were comparing (previously listed in Table I) didn't consider $\theta_{\text{FL},x}$ or $\theta_{\text{FL},z}$ either.

Second, the SHH equation in Supplementary Eq. (2) shows that $R_{\text{FL},z}$ is a constant term, which could be more easily affected by offset signals, et al. Therefore, we don't think the calculated $\theta_{\text{FL},x}$ or $\theta_{\text{FL},z}$ will give reliable values of the FL spin torque efficiency.

f. In the ST-FMR section of methods, the model does not consider other spin polarization so the derived spin torque efficiency cannot make direct comparison to the SHH results.

From Fig. r1-4, $\tau_{AD,x}$ could be mixed with $\tau_{AD,y}$ when calculating θ_{AD} using ST-FMR measurement. Therefore, the spin torque efficiency achieved from ST-FMR is actually from a combination of x-polarized spin and y-polarized spin. From the data obtained using SHH measurement, these samples sputtered on Si/SiO₂ substrates have unconventional spin torque efficiency from x-polarized spin less than 20% of the conventional one from y-polarized spin. Therefore, we allow for some differences between the two methods. As the reviewer suggested, we should not directly compare the two numbers but clarify the difference. We have revised the statement (**Lines 197-199** in the revised manuscript) to clarify the difference and explained the consistency between the two measurements.

Fig. r1-4 A schematic of spin torques in ST-FMR measurement.

g. Line 362, there is no indicative trend of “close to saturation”.

Thanks much for pointing this out. We agree this description is not accurate or clear. Accordingly, we have revised the paragraph (**Lines 347-351** in the revised manuscript), in which we now emphasize that the thickness-dependent experimental data (Fig. 7c) follows the trend of magnetization-dependent spin Hall conductivity σ^{yz} (Supplementary Fig. 3) calculated with first-principles method from MSHE.

h. Line 373, how to understand “The ultrathin Co multilayer in a paramagnetic state is likely to be at the ideal magnetization orientation close to 45deg in the xy-plane.” The paramagnetism and ferromagnetism co-exist?

We agree the original sentence is misleading. What we mean is that the ultrathin Co multilayer is paramagnetic rather than ferromagnetic, but it gets weakly magnetized in the presence of a magnetic field. Accordingly, we have revised the sentences (**Lines 425-428** in the revised manuscript) to emphasize that the weak Co magnetization may be induced by the stray magnetic field from the in-plane magnetized CoFeB and the field-like torque effective-field from the conventional spin polarizations. We have also removed the “45 degree in the xy-plane” which is possible but not experimentally confirmed.

i. Line 261, how to understand “Spin currents J_{σ_x} and J_{σ_y} both flow in z-direction and exert torques on magnetization M with σ_y breaking the inversion symmetry.”

We agree that this sentence is not clear, so we have rewritten the sentences (**Lines 269-271** in the revised manuscript) to clarify two points we’d like to convey through Fig. 5d, which is in the type-y SOT switching configuration: Both σ_x and σ_y spin currents can exert torques on the CoFeB layer and they are of similar magnitude and opposite sign ($\theta_{AD,x} = -0.083$ and $\theta_{AD,y} = 0.102$); The σ_y spin polarized in a uniaxial direction is parallel with the magnetic easy axis so that it can break the symmetry and switch FM layer deterministically in type-y configuration. We have also revised the parallel sentences (**Lines 252-258** in the revised manuscript) which describes Fig. 5a about the type-x SOT switching configuration. In addition, we have redrawn the schematic plots of Fig. 5a and 5d (**Lines 283** in the revised manuscript).

2. The schematic of stacking all three MgO vertically in Fig. 1a is inappropriate and misleading. And for a better logic flow, Fig. 1 d,e should be moved to the position before Fig. 1 b,c.

Thanks much for the great suggestion, we have separated the three MgO crystalline structures in Fig. 1a and reorganized the plots in Fig. 1 (**Lines 126** in the revised manuscript). The context (**Lines 107-109 & 114-117** in the revised manuscript) and figure citations (**Lines 129-131** in the revised manuscript) have been modified accordingly.

3. SRT in Fig. 1d,e and pure Pt sample in line 213 are not defined beforehand.

Thank you for pointing out, we have added a definition of SRT and the sample stack abbreviation (**Lines 97-98 & 149-150** in the revised manuscript) when first presenting it. This “pure Pt-based sample” has also been rephrased (**Line 230** in the revised manuscript) to clarify its stack as a reference sample.

4. Table 1 makes little meaning as the prior results are too few. It may provide wrong indication of large unconventional spin polarization in this work.

The original intention of Table I was to place this work in the context of the commonly used or reported SOT materials for high energy-efficiency and field-free SOT switching. We want to add two notes here about the table: 1) not all reported SOT materials were investigated with both x-polarized spins and z-polarized spins in the references; 2) our reported unconventional x-polarized spin torque efficiency is so far the largest among the known reports. However, we agree that the table is not comprehensive in the regard of comparing all emerging SOT materials. Therefore, in our revised manuscript, we have referred the comparison to a well-summarized table in Ref [15] and have replaced the table with descriptive sentences (**Lines 121-125** in the revised manuscript).

5. The coercivity of the SRT1 and SRT2 sample are only few Oe which make the switching experiment very suspicious. The switching could be related to the current induced Oersted field or the STT effect between [Pt/Co]_n and CoFeB etc.

Thanks for the question! Even though the coercivity is quite small (~5 Oe) in the Hall-bar devices, we do believe SOT switching is observed in the SRT1/Mg/CFB stack in the absence of external magnetic field. Besides the explanation in the original manuscript, we have added some sentences (**Lines 263-266** in the revised manuscript) to address the concerns from the reviewer’s comments. (Note that Fig. 5 is only about SRT1/Mg/CFB samples for type-x and type-y switching in absence of external fields during write. The SRT2-based samples don’t display field-free type-x switching as implied in your comment.)

- 1) The current-induced Oersted field is along the y direction when the charge current is along the x direction, according to Ampere's Law. For type-x configuration, a magnetic field along the y direction cannot break the inversion symmetry to deterministically switch magnetization between +x and -x. In addition, the reference sample shown in Fig. 4 didn't show any field-free switching from the current-induced Oersted field. Together, these considerations strongly support that the full switching in type-x configuration is not due to current-induced Oersted field.
- 2) The STT effect is negligible in Fig. 5 because the current is applied in in-plane direction so that no net charge current is going through the film stack in perpendicular direction. Furthermore, the ultrathin Co in SRT1 is paramagnetic and weakly magnetized, so its spin polarization, if any, is tiny, and the resulting STT effect from Co to CoFeB will be negligible.

6. At the end, the authors should try to make more concise presentation and discussion. One will be confused by the MgO orientation dependence, the Co thickness dependence, the SAHE/SHE/MSHE/MOHE mechanisms etc. They are very much out of organization.

We understand that the previous version is not presented clearly and not easy to read. We have revised the corresponding sections (**Lines 337-442** in the revised manuscript) to make them more logical and coherent. A brief summary of the storyline is given here: This manuscript includes a comprehensive study of $[\text{Pt}/\text{Co}]_n$ stacks as a spin current source by generating both conventional and unconventional polarized spins. In the beginning of the paper, we introduced a large unconventional spin torque efficiency discovered in both textured and amorphous substrates. After presenting the significant x-polarized spin torque efficiency from stacks deposited on (100) MgO and (110) MgO substrates, we then show the x-polarized spin torque efficiency in stacks deposited on the amorphous surface of a silicon substrate (Si/SiO_2), which is applicable to MRAM products in industry. Subsequently, we present the SOT performance of the samples on Si/SiO_2 , including the Co-thickness-dependent measurement. To explain the generation of unconventional spin polarizations, we conduct XMCD experiments which have revealed strong orbital moment in low-dimensional Co. This leads to stronger spin-orbital scattering so that enhanced spin torque efficiency, especially unconventional spin polarizations due to low symmetry, is observed. We

alternatively use the MSHE from the macroscopic perspective to explain the Co-thickness-dependent trend in the experiment.

We have reorganized the data presentation and explanation in the revised manuscript to clarify the mechanisms of unconventional spin generation. The part with MOHE-related explanation has been removed. We appreciate the reviewer's suggestions and comments, and hopefully we have addressed all your concerns!

Reviewer #2 (Remarks to the Author):

Xue and colleagues report investigations of current-induced switching of type-x SOT devices with [Pt/Co]_N underlayers. Type-X SOT devices attract increasing attention owing to high-speed switching. Compared to type-y devices, the channel width of type-x can be reduced, leading to a relatively small switching current. However, an out-of-plane external field is required to induce deterministic switching in conventional structures. The results of the harmonic Hall measurement show the unconventional spin current, which is corresponding to the x-component of spin polarization. Using this unconventional spin current, authors achieve field-free SOT switching in type-x devices. The XMCD measurement indicates magnetic anisotropy of Co layer which provides a possible mechanism of the unconventional spin current. This work is a high-quality experimental study and may be of interest to a broad community but I encourage the authors to address the following comments before the publication.

Thanks so much for your recognition and great support for our work. *We have carefully addressed your comments point-by-point and polish the manuscript to the best of our efforts.* Revisions have been highlighted in the revised manuscript and Supplementary Information. We hope this revised version is good to publish.

1. In Fig.1(b), the current direction dependence of spin torque efficiency due to x-spin polarization ($\theta_{AD,x}$) is displayed and large efficiency is observed in stacks on MgO(100) and (110) when $J // 90\text{deg}$. However, for stacks on MgO(100) substrate, $J // 0\text{ deg}$ and $J // 90\text{ deg}$ indicate the equivalent experimental configuration and similar spin torque efficiency is expected. I suggest that the authors discuss this point.

In the mechanism of crystalline-related x-polarized spin generation, the charge current direction is crucial for inversion symmetry breaking. Taking together Fig. 1a and Fig. 1d in the revised manuscript, it's only when charge current along [001] generates large unconventional spin torque efficiency $\theta_{AD,x} \sim -0.083$; in other scenarios, $\theta_{AD,x}$ keeps around -0.02 from the experimental data. For stacks deposited on (100) MgO, $J//0^\circ$ is along [010] direction and $J//90^\circ$ is along [001] direction. These two directions will indicate equivalent configurations only in the scenario of out-

of-plane texture dominating spin generation. However, as shown from the 2-D XRD measurement in Fig. 2b, we didn't find any specific out-of-plane texture in SRT1 stack that could be responsible for high $\theta_{AD,x}$. Therefore, as both of the SHH data and XRD data imply, an in-plane texture could be dominant for symmetry breaking. We speculate that the in-plane crystalline asymmetry is resulted from the magnetocrystalline anisotropy induced from the magnetic-field distorted plasma during magnetron sputtering. For example, Uchiyama et al. [IEEE Trans. Magn. 26, 5 (1990)] used magnetic fields during sputtering to modulate the microstrain and lattice coherency in the CoCr film for coercivity modification. The paragraph has been revised accordingly to be clear (**Lines 114-121** in the revised manuscript).

2. There is a paper reporting the SOT-induced switching of in-plane magnetized CoFeB nanodots with various angles (ϕ_{EA}) between the magnetic easy axis and current direction [Takahashi et al., Appl. Phys. Lett. 114, 012410 (2019)]. The previous paper demonstrates field-free SOT switching above ϕ_{EA} of 10 deg by using pulse current of 1.7 ns – 100 ms. I understand that the mechanisms of SOT are different between the submitted manuscript and the previous study, however, the switching property such as threshold current vs pulse width might not be largely different. I propose that the authors compare this previous study with the submitted manuscript and discuss the difference in switching properties.

Thanks so much for sharing this paper with us. Our submitted manuscript has a good consistency with this APL paper from Takahashi et al. Since the reviewer is asking about threshold current vs. pulse width, we believe the question is about the micromagnetic simulation in our manuscript shown in Fig. 6. Here, we compare the two papers and explain the differences.

- 1) Switching mechanism: The two papers are actually of the same switching mechanism for field-free switching, which is to switch the type-x configuration through x-polarized spin plus y-polarized spin. The difference occurs in the origins of x-polarized spins. In Takahashi paper, the coordinate system is based on the Hall-bar structure, and the x-polarized spins are generated by in-plane tilted magnetic shape anisotropy of FM layer. In Xue paper, the x-polarized spins are generated by the SOT layer with symmetry breaking. The cases of $\phi_{EA}=10^\circ$ and 80° [Fig. 3b, Takahashi paper] respectively approximate the

situations of type-x ($\theta_{AD,x}/\theta_{AD,y} = 0.10$) and type-y ($\theta_{AD,x}/\theta_{AD,y} = 0.10$) switching [green solid and green open circles in Fig. 6a, Xue paper].

- 2) Measurement method: They both use differential planar Hall effect (DPHE) for in-plane magnetization switching measurement. The offset (or bias) field for DPHE measurement is 50 Oe [Takahashi paper] and 3.2 Oe [Xue paper], respectively.
- 3) Threshold current density vs. pulse width: The dimension of CoFeB free layer is $100 \times 400 \times 1.68 \text{ nm}^3$ [Takahashi paper] and $128 \times 384 \times 2.5 \text{ nm}^3$ [Xue paper]. The calculated H_k is close to each other. Furthermore, considering the rounded edge (ellipsoidal shape) of the fabricated magnetic cell, the larger spin Hall angle in Ta(7)/W(1.6) than in Pt, and the reduced H_k by larger offset field of 50 Oe in DPHE measurement, the switching current density in Takahashi paper could be at least one order smaller than the simulation data in Xue paper. In addition, Takahashi paper tests devices in room temperature at $\sim 300 \text{ K}$ and Xue paper simulates at 0 K . At 0 K without thermal fluctuation, the threshold current density is larger and the pulse width gets smaller. Qualitatively comparing, the case of $\varphi_{EA} = 10^\circ$ or type-x ($\theta_{AD,x}/\theta_{AD,y} = 0.10$) switching shows larger current density than that of $\varphi_{EA} = 80^\circ$ or type-y ($\theta_{AD,x}/\theta_{AD,y} = 0.10$) switching, which is consistent in the two papers.

We referred to this paper (Lines 318-322 in the revised manuscript) in the simulation section to present the consistency between simulation and experiment.

Fig. r2-1 referring to Fig. 3b [Takahashi paper] (left) and Fig. 6a [Xue paper] (right)

3. In figure 2, out-of-plane R_H - H curves and angle dependence of harmonic Hall resistance are displayed. Because Co/Pt multilayer of SRT1 sample is close to a paramagnetic state, the anomalous Hall resistance of CoFeB layer is dominant in Fig.2(c), and the result of the 2nd harmonic Hall resistance (Fig.2g) reflects the magnitude of SOT acting on CoFeB. Meanwhile, in SRT2 sample, the Hall resistance is mainly due to Co/Pt multilayer, therefore, the result of Fig.2h may be originating from the SOT on Co/Pt layer. Before the comparison of SOT efficiency as shown in Fig.7c, I would suggest that the authors extract the magnitude of SOT acting on CoFeB layer even for SRT2 sample.

Thanks much for this question. We agree that the SHH measurement includes signals from both SRT2 layer and the in-plane CoFeB layer. In Fig. r2-2, we showed the data of $R_{H,1\omega}$ vs. φ and $R_{H,2\omega}$ vs. φ from angular-dependent SHH measurement, tested on the SRT1 sample and the SRT1/Mg/CFB sample (Fig. r2-2a and 2b), the SRT2 sample and the SRT2/Mg/CFB sample (Fig. r2-2c & 2d) deposited on Si/SiO₂.

Fig. r2-2 Angular-dependent SHH measurement: (a) (c) $R_{H,1\omega}$ vs. φ and (b) (d) $R_{H,2\omega}$ vs. φ at $H_{\text{ext}}=200$ mT.

The data of $R_{H,1\omega}$ and $R_{H,2\omega}$ in the SRT1 samples is negligible comparing to that of the SRT1/Mg/CFB sample because SRT1 is in paramagnetic state and is only weakly magnetized. The magnitude of $R_{H,1\omega}$ is of similar order in the SRT2 and SRT2/Mg/CFB samples; while $R_{H,2\omega}$ is negligible in SRT2 comparing to that in SRT2/Mg/CFB. In terms of the data shown in Fig. 2 in the manuscript, we didn't do subtraction to calculate the torques acting on CoFeB layer due to:

- 1) In $R_{H,2\omega}$, signal from the $[\text{Pt/Co}]_n$ stacks is quite negligible comparing to the $[\text{Pt/Co}]_n/\text{Mg/CFB}$ stacks, no matter it is SRT1 or SRT2.
- 2) From the perspective of equivalent circuit, the $[\text{Pt/Co}]_n$ stack and the CFB layer are in parallel-resistance model; however, it is not correct to claim $1/R_{H,[\text{Pt/Co}]/\text{Mg/CFB}} = 1/R_{H,[\text{Pt/Co}]} + 1/R_{H,\text{CFB}}$ for either $R_{H,1\omega}$ or $R_{H,2\omega}$.
- 3) The two FM layers as of $[\text{Pt/Co}]_n$ and CFB are not in superposition, either. Simply subtracting one from the whole stack doesn't make any meaningful results.
- 4) We didn't see references with double-FM systems providing reasonable approach to calculate efficiencies by extracting SHH signal from one layer. The ST-FMR measurement could probably solve this issue due to different resonant fields of different FMs if decoupled.

Therefore, we calculated SOT efficiencies from SHH signals of the whole stacks. We have notated the negligible $R_{H,2\omega}$ from $[\text{Pt/Co}]_n$ stack (**Lines 169-170** in the revised manuscript), and added this figure into Supplementary Figures (**Lines 216-219** in the revised Supplementary Information).

4. On page 19, it is mentioned that the magnetization orientation is close to 45 deg away from the film normal direction. I think that the 45 deg tilt of magnetization is not very convincing because it requires another different magnetic anisotropy rather than both perpendicular and in-plane uniaxial magnetic anisotropies. Experimental evidence for such unconventional magnetic anisotropy is needed. Otherwise, the discussion part of the submitted manuscript needs to be toned down.

We agree that the original statement is too strong if without experimental data to support. What we mean is that the ultrathin Co multilayer is paramagnetic rather than ferromagnetic, but it gets weakly magnetized in the presence of a magnetic field. Accordingly, we have revised the

sentences (**Lines 425-428** in the revised manuscript), and have also removed the “45 degree in the xy-plane” which is possible but not experimentally confirmed.

5. In line 386 of page 20, ref 34 is cited as a report on the self-switching of perpendicular magnetized CoPt. But ref 34 reports the manipulation of exchange bias by SOT in Pt/Co/IrMn. The citation part should be corrected.

Thanks much for pointing this out. We accidentally mixed references [18] and [34] in the original submission, and we have swapped them back in the revised manuscript (**References [10] and [32]** in the revised manuscript). All other references have also been double checked and reorganized according to the revised manuscript.

Reviewer #3 (Remarks to the Author):

Xue et.al. reported the field-free switching of type x in-plane magnetization by pure electric current which is paramount important for SOT-MRAM application with fast switching since type x in-plane shows more than 10 times faster magnetization switching induced by electric current than those of type y in-plane and type z perpendicular magnetization. Particularly, the materials structure used in the manuscript is compatible with industry fabrication process. Fundamentally, they found an unconventional spin current with spin polarization in x direction in a simple material-ultrathin Co layer in Co/Pt multilayer, which is demonstrated by the detailed electric transport and material characterization. Spin current with x spin polarization is attributed to the strong spin orbital coupling of ultra-thin Co layer, which has never been observed before. The presented results are both fundamentally and technically interesting, especially important in the fields of spintronics and condensed matter physics. Therefore I would recommend it for publication in Nature Communication after they address the following technical comments.

Thanks so much for your nice summary of our work and support of the paper publication. *We have addressed the technical comments point-by-point, and revised accordingly in the manuscript.* Revisions have been highlighted in the manuscript. We hope this revised version is much more improved and can get your approval on publication.

1) The figure 1 schematically shows relationship between crystal direction and current direction. It is confused that the experiment seems not consistent with this drawing. e.g. how electric current // [001] direction is applied? It is quite usual that one use [001] direction to refer to film normal. In Fig.1 b, $J//90^\circ$ for the films grown on Si/SiO₂, MgO (111) are different from MgO(100), MgO(110), it is easy to cause the misleading. Same for $J//45^\circ$. In addition, Co layer grown on MgO single crystal substrate should be fcc structure, therefore, [001] and [010] should be equivalent, why the AD_x and AD_y are different for the sample on MgO(100) substrate?

We are sorry that the original description for Fig. 1a might be not adequate or clear enough about the charge current direction and the crystalline direction. We have re-drawn Fig. 1a (**Line**

126 in the revised manuscript) and revised the figure citation (**Lines 129-133** in the revised manuscript) to clarify the angle of charge current and its direction in the crystalline coordinate.

- 1) To answer the questions, (100), (110), and (111) represent three plane directions, as indicated by the shadowed planes in Fig. 1a. For example in the (100) MgO substrate, [100] is the film normal direction, and $\mathbf{J} // [001]$ represents \mathbf{J} along the in-plane 90° .
- 2) In Fig. 1d in the revised manuscript, the data points in the curve of $\mathbf{J} // 90^\circ$ on different substrates don't represent the same charge current direction in the crystalline coordinate. For example, in the curve of $\mathbf{J} // 90^\circ$, the charge current direction in the crystalline coordinate is respectively [001], [001], and $[11\sqrt{2}]$ on the substrates of (100) MgO, (110) MgO, and (111) MgO. Similar to $\mathbf{J} // 0^\circ$ and $\mathbf{J} // 45^\circ$. As indicated in Fig. 1a and its figure citation (**Line 130-132** in the revised manuscript), there are seven charge current directions we have applied in the crystalline coordinate. We can choose to plot the data in the way of efficiency vs. crystalline direction (e.g. [001], [011], [010], ...), then it will show 2 data pts in [001], 2 data pts in [110], 1 data pt in [011], [010], et al. As a comparison, we prefer to present our data in its original form which looks more organized.
- 3) We concur that Co typically exhibits a fcc structure when deposited on MgO single crystals. The difference of SOT efficiency when charge current along the [001] and the [010] directions can be attributed to the in-plane texture. We speculate that this in-plane crystalline asymmetry results from the magnetocrystalline anisotropy induced by magnetic-field distorted plasma during magnetron sputtering. The plasma can be distorted by magnets that rotate along with the sample holder, leading to asymmetry and lattice strain in the film deposition and crystallization. Uchiyama et al. [IEEE Trans. Magn. 26, 5 (1990)] utilized a magnetic field during sputtering to modulate the microstrain and lattice coherency in a CoCr film, effectively controlling coercivity.

2) Do the samples SRT1 and SRT2 any shape anisotropy? In Fig.2c, which is the direction of applied field, along current direction or transverse to current direction? since it is measured for the whole stack, CFB in figure 2c is also superparamagnetic?

In Fig. 2c and 2d, H_{in} refers to H_x , which is aligned with the charge current direction. After fabricating the film stacks into Hall-bar devices with a $2\mu\text{m}$ -width current channel, shape

anisotropy was formed. However, since the micro-scale device is domain-wall dominated, the magnetic anisotropy is primarily induced by the magnetic field during sputtering, rather than the shape anisotropy resulting from fabrication. The in-plane coercivity is quite small, and we did not display the back-and-forth full loop of R_H vs. H_{in} .

Figure 2c shows that CFB is not paramagnetic but requires a larger in-plane field to saturate the R_H value. We also note that the R-H curves from Hall measurement do not correspond directly to M-H curves. The in-plane field generated by our magnetic coil is limited to ~ 7.5 kOe due to the constraints of the current source and the space between the two coils.

3) “The asymmetry of switching current from +x to -x and -x to +x originates from the asymmetric coercivity due to unbalanced energy states of domains in +x and -x without annealing.” Why there exists unbalanced energy state? What means “without annealing”? do you mean that after annealing, there will be unbalanced energy state?

We acknowledge that the “unbalanced energy states” used in the original submission could be misleading to readers. From the field-switch data, we observed that the coercivity is not symmetric between the negative and positive branches. After ruling out the effect of exchange bias (due to the absence of AFM materials or coupling in the SRT1/Mg/CFB stacks and variation in the asymmetry across different samples), it would be more appropriate to describe this phenomenon as “asymmetry”. Several factors could contribute to the magnetization asymmetry along the +x and -x directions, including asymmetric domain nucleation [Nikitenko, et al., Phys. Rev. B 57, 14 (1998)], rough edge pinning effects [Deak, et al., J. Magn. Magn.Mater. 213 (2000)], asymmetric magnetocrystalline energy induced by the magnetic fields during magnetron sputtering [Uchiyama et al., IEEE Trans. Magn. 26, 5, (1990)], random domain orientations due to surface roughness, or asymmetric dimension resulting from patterning. Post-anneal can help to reduce some of the asymmetry. We have briefly revised **Lines 261-262** in the revised manuscript to clarify the reasoning of asymmetry.

4) I noticed that the authors stated “It should be noted that the small bias field H_y is for the purpose of DPHE measurement alone and is not able to break the symmetry of the type-x SOT switching.”

Do they apply a bias field during switching field? If yes, authors may try during switching, do not apply the bias field. Only applying the bias field to distinguish the magnetization direction after switching.

Thank you for your insightful question. As depicted in Fig. 4c, the small bias field is only present during the Hall resistance measurement (read state), which means we don't need to consider any external field during current-induced magnetization switching. We realized that the original sentence may have caused confusion, so we have added a clarification in **Line 226** in the revised manuscript. From another angle, even there is external field present along the y-direction during writing, it doesn't break the symmetry in the type-x configuration, but only reduces the energy barrier for magnetization reversal.

5) In Fig.7, the authors used wedge samples which will introduce the asymmetry in the wedge direction, which may induce the different spin polarization. It would be good not use the wedge sample to evaluate the thickness dependence on the different spin polarization induced AD torque.

Thank you for the insightful question and suggestions. We agree that wedged samples may induce different polarized spins due to asymmetric dimension. Therefore, we investigated the thickness-dependent magnetization and efficiencies on wedged-samples first, and determined the optimal thickness for unconventional polarized spins. Subsequently, we sputtered single samples at a certain fixed thickness to double-check the SHH signals and conduct field-free switching, as shown in Fig. 1-5 and 8.

From another perspective, we don't think that the wedged structure had a significant effect on our results for two reasons: 1) When patterning on the wedged samples, we made sure that the wedged direction was transverse to x (current direction) to avoid symmetry breaking from lateral structure. 2) The thickness gradient is only $\sim 1.55\%/mm$. For a $10\ \mu\text{m}$ -width current channel and $0.28\ \text{nm}$ -thickness at the center of the wedged-sample (Fig. 7c), the thickness change is only 155 ppm ($43.4\ \text{fm}$) within a device.

Furthermore, the data obtained from the wedged-samples and single-thickness samples are consistent, which is within the range of device variations.

Reviewers' Comments:

Reviewer #1:

Remarks to the Author:

The authors have devoted great efforts to address the concerns raised by the referees. However, I find the responses and revisited version are not satisfactory to reach the publication criteria of Nat. Commun. The whole manuscript is still chaotic in organization, making one difficult to read and catch the idea. The figures are too many and at least 2~3 of the can be moved to the supplementary as they are not really relevant to the central subject. The "type-x" in title is too specific and even might be misleading. The type x is not attractive for practical application due to its low nonvolatile ability. The proposed underlying mechanism is also not convincing. Taking all these critical concerns all together, I do not recommend its acceptance in Nat. Commun.

Reviewer #2:

Remarks to the Author:

The authors have addressed all the points raised in the previous review report and I recommend now the paper for publication.

Reviewer #3:

Remarks to the Author:

The authors have addressed my concerns in their response and revised manuscript. I recommend it for publication.

Reviewer #1 (Remarks to the Author):

The authors have devoted great efforts to address the concerns raised by the referees. However, I find the responses and revisited version are not satisfactory to reach the publication criteria of Nat. Commun. The whole manuscript is still chaotic in organization, making one difficult to read and catch the idea. The figures are too many and at least 2~3 of the can be moved to the supplementary as they are not really relevant to the central subject. The “type-x” in title is too specific and even might be misleading. The type x is not attractive for practical application due to its low nonvolatile ability. The proposed underlying mechanism is also not convincing. Taking all these critical concerns all together, I do not recommend its acceptance in Nat. Commun.

Thanks much for your feedback with critical thinking on our revised manuscript. We believe the most updated version has clearly illustrated the presence of significant unconventional x spin polarizations in ultrathin $[\text{Pt}/\text{Co}]_N$ and its assist in field-free type- x SOT switching. We value your input very much, so we are trying to give some more explanations and revisions from new aspects, and hopefully it is convincing from the perspective of a reviewer. Here, we will *address the five comments point-by-point as shown in our review response*. Revisions are highlighted in the manuscript.

1. This manuscript is organized logically as follows. Following the Introduction section, we first presented the significant unconventional spin polarizations by material characterizations. Next, we demonstrated the unconventional-spin-assisted field-free switching at device level. Then, we explored the dynamics of x -spin-assisted switching using micromagnetic simulations. Finally, we discussed the Co magnetization and thickness dependence of spin polarizations, and elucidated the origin of unconventional spin polarizations generated in low-dimensional Co using XMCD measurement. We have inserted subheadings in the Results section to guide the readers.
2. With regard to the number of figures, we have made revisions to the main manuscript to improve the clarity of the storyline. Here are the changes we have made: We have moved Fig. 3, which presents the SOT characterization results using independent measurements, to the Supplementary Fig. 2. Figure 4, which includes the description of DPHE measurement and results from a reference sample, has also been moved to the Supplementary Fig. 3. By

removing these two figures from the main manuscript, we have maintained the logical flow and overall completeness of the document. The remaining figures in the manuscript have been retained to ensure a smooth and comprehensive presentation of our work. The necessary adjustments have been made in the corresponding sections of the main manuscript.

3. We have made a revision to the title by removing the term “type-x” to make it more applicable to various applications and increase its appeal to a wider range of readers. Since our work is more focusing on type-x SOT switching, we have reflected this emphasis in the abstract.
4. With regard to the nonvolatile ability, the metrics to describe it is the thermal stability ratio Δ . Δ in type-x can be similar to that in type-z; additionally, as MTJ size scales down, Δ in type-x can maintain and show smaller switching current than in type-z. We give the explanations as below and put the simulation data in Supplementary Fig. 1 to demonstrate our statement concretely.

In a single-domain model, $\Delta = \mu_0 H_k M_s V / (2k_B T)$, where H_k is the magnetic anisotropy of the free layer (FL), and V is its volume. The magnetic shape anisotropy is $H_{k,IMA} = 2M_s t (AR - 1) / (wAR)$ in in-plane magnetic anisotropy (IMA), and $H_{k,PMA} = 2K_u / (\mu_0 M_s) - M_s$ due to interfacial energy in perpendicular magnetic anisotropy (PMA). Here, t , w , and AR are the thickness, width, and aspect ratio of FL, respectively; K_u is the interfacial magnetic energy constant. To compare the H_k , we can make a quick calculation here. Taking CoFeB as the FL, assuming MTJ CD=18 nm, $AR=3$, $t_{IMA}=2$ nm, $t_{PMA}=1$ nm, and $K_u=1$ MJ/m³, $H_{k,IMA}$ is ~ 2.6 kOe and $H_{k,PMA} \sim 15$ kOe. Considering the in-plane magneto-crystalline anisotropy in IMA ($K_{u,in} \sim 40$ kJ/m³), $H_{k,IMA}$ can be adjusted to ~ 3.2 kOe. Due to the volume benefit in type-x which is with a factor of $AR * t_{IMA} / t_{PMA}$, Δ_{IMA} is about 1.3 times of Δ_{PMA} . Further, type-x is easier to maintain high Δ than type-z when scaling down due to increasing H_k and flexible FL thickness.

In the micromagnetic simulation, we used a fixed thermal stability target of $60 k_B T$ at 350 K with a current pulse width of 0.8 ns. We found that the external field requirement H_x in type-z MTJs increases as the MTJ critical dimension (CD) decreases. In order to enable deterministic switching at all MTJ CDs considered, we select a constant external field of 120 mT for both type-x and type-z MTJs and observe how the critical current changes as a function

of MTJ CD, shown in Fig. r1-1. We found that the critical switching current increases as the MTJ CD decreases in type-z MTJs, whereas the critical switching current remains relatively constant in type-x MTJs. More specifically, the switching current in type-x MTJs becomes more favorable than type-z MTJs at CDs of ~ 30 nm or below, owing to the lower anisotropy field requirement and higher free layer volume available in x-type MTJs.

Fig. r1-1 Critical switching current in type-x and type-z SOT configurations dependent on MTJ CD. In both configurations, external field is 120 mT, thermal stability ratio is 60 at 350 K, the current pulse width is of 0.8 ns. In the type-x MTJ cell, aspect ratio (AR) is 3; in the type-z MTJ cell, the free layer (FL) thickness is set at 1 nm.

In summary, type-x configuration can be very attractive in SOT switching due to reasons below:

- 1) At the same thermal stability ratio, type-x needs smaller current and current density when scaling down, e.g. MTJ CD at smaller than 30 nm. Or put it in another way, retention of type-x SOT-MRAM can be higher than that of type-z SOT-MRAM in deep technology nodes where MTJ CD is smaller than 30 nm.
- 2) Much smaller unconventional spin polarization efficiency is required for field-free switching in type-x than in type-z, referring to Lin et al. 2021 IRPS.
- 3) Potentially less spin precession and faster switching in type-x SOT switching is needed than in type-z SOT switching due to demagnetization which is also demonstrated in micromagnetic simulation in Fig. r1-2, which has been shown and described explicitly in last round of response.

Fig. r1-2 Dynamic magnetization trajectories of type-x switching (a)-(c), and type-z switching (d)-(f). Current density, external field, and spin polarizations have been indicated in the plots.

- To make the mechanism more convincing in explaining the presence of significant unconventional polarized spins in the weakly in-plane magnetized Co films, we have added statements in the discussion section, and a symmetry-breaking illustration in Supplementary Fig. 6, as shown below (Fig. r1-3). With current applying along c -axis and under magnetic mirror symmetry by \mathcal{M}' , the unconventional spin-orbit torque τ_x is allowed. This indicates the symmetry breaking by in-plane magnetization is highly likely to enable unconventional spin polarization generation in this system. Along with support from other publications (cited in the main manuscript) on the unconventional spin generation mechanism in low-symmetry materials, we believe we have made a strong qualitative argument in this study. We sincerely wish that our manuscript will stimulate more studies on this nascent spintronics topic.

Fig. r1-3 An illustration showing the symmetry breaking in weakly in-plane magnetized Co films. The bc -plane is the in-plane of the film, current I is applying along the c -axis, and the unconventional spin-orbit torque τ_x from x -polarized spins is along the c -axis, too. The green spheres and red arrows represent the Co atoms and magnetic moment, respectively. The blue dashed line represents the magnetic mirror symmetry \mathcal{M}' , under which, the unconventional spin-orbit torque τ_x is allowed.

Reviewer #2 (Remarks to the Author):

The authors have addressed all the points raised in the previous review report and I recommend now the paper for publication.

Thank you for your positive assessment of our revised manuscript. We greatly appreciate your time and effort in reviewing our work. We are pleased to hear that you found that we have adequately addressed all the points raised in the previous review report. Based on your recommendation for publication, we are delighted that our manuscript has met the standards of the journal and is now ready to be shared with the scientific community. We are grateful for your recognition of our efforts and the confidence you have expressed in the quality of our work.

Once again, we extend our sincere gratitude for your valuable feedback and guidance throughout the review process. Your contributions have undoubtedly played a crucial role in shaping our manuscript and making it ready for publication. Thank you once again for your time, support, and recommendation.

Reviewer #3 (Remarks to the Author):

The authors have addressed my concerns in their response and revised manuscript. I recommend it for publication.

We sincerely appreciate your careful evaluation of our revised manuscript and for acknowledging that we have adequately addressed your concerns in both our response and the revised version of the paper. Your feedback and recommendations have been invaluable in improving the quality and clarity of our work. We are delighted to hear that you now recommend our manuscript for publication. Your positive assessment further validates our efforts and gives us confidence that our research will make a valuable contribution to the scientific community.

Thank you once again for your time, constructive feedback, and for recommending our paper for publication. We greatly appreciate your support throughout the review process.